# A gene regulatory network to control EMT programs in development and disease

Hassan Fazilaty [1,2], Luciano Rago [1,3], Khalil Kass Youssef [1], Oscar H. Ocaña [1], Francisco Garcia-Asencio [1], Aida Arcas [1,4], Juan Galceran [1] & M. Angela Nieto [1]*

The Epithelial to Mesenchymal Transition (EMT) regulates cell plasticity during embryonic development and in disease. It is dynamically orchestrated by transcription factors (EMT-TFs), including Snail, Zeb, Twist and Prrx, all activated by TGF-β among other signals. Here we find that Snail1 and Prrx1, which respectively associate with gain or loss of stem-like properties and with bad or good prognosis in cancer patients, are expressed in complementary patterns during vertebrate development and in cancer. We show that this complementarity is established through a feedback loop in which Snail1 directly represses *Prrx1*, and Prrx1, through direct activation of the miR-15 family, attenuates the expression of Snail1. We also describe how this gene regulatory network can establish a hierarchical temporal expression of Snail1 and Prrx1 during EMT and validate its existence in vitro and in vivo, providing a mechanism to switch and select different EMT programs with important implications in development and disease.

[1] Instituto de Neurociencias (CSIC-UMH), Avda. Ramón y Cajal s/n, Sant Joan d', Alacant 03550, Spain. [2] Present address: Institute of Molecular Life Sciences, University of Zurich, 8057 Zurich, Switzerland. [3] Present address: Department of Oncogenomics, Academic Medical Center, Amsterdam, the Netherlands. [4] Present address: Department of Gene Therapy and Regulation of Gene Expression, Center for Applied Medical Research, University of Navarra, Pamplona, Spain. *email: anieto@umh.es

The epithelial to mesenchymal transition (EMT) is a developmental process that can be ectopically reactivated in diseases like cancer. When they undergo EMT, epithelial immotile cells become mesenchymal, acquiring the ability to migrate and invade[1]. During tumor progression, EMT provides invasive and migratory properties to cancer cells[1–4]. This phenotypic transition is governed by extracellular signals that activate a plethora of EMT transcription factors (EMT-TFs) that include those belonging to the Snail, Zeb, Twist, and Prrx families[1]. Different mesenchymal cells of embryonic, healthy adult or pathological origin, express different combinations of EMT-TFs, leading to a tissue-specific EMT-TF code[5], which may influence the overall state, function, and behavior of the cell. In addition to the transition toward the mesenchymal phenotypes, Snail1, Twist1, and Zeb1 can also induce stemness[6,7], while Prrx1 expression is concomitant with the loss of stemness[8,9]. In cancer, highSnail1 expression is associated with malignant phenotype and poor prognosis[10,11], while the high expression of Prrx1 is associated with good prognosis and metastasis-free disease[8]. In the chicken embryo, *PRRX1* and *SNAIL1* are expressed in a complementary manner[8] and in breast cancer Prrx1 expression correlates with that of Twist1 but not Snail1[8]. These differences can be considered as different EMT modes associated with the dominant EMT-TF in a given cellular context[5]. Studying the differences between all these EMT-TFs is important to understand cell plasticity during embryonic development, which can ultimately help to distinguish the key altered cellular and molecular mechanisms in disease.

Combined expression of *SNAIL1* and *PRRX1* covers almost the entire mesenchymal cell population in the chicken embryo[8]. Although there are clear differences in the EMT activated by each factor in development and cancer, the two are activated by the same extracellular signals, the transforming growth factor beta (TGF-β) superfamily[8,12]. Therefore, we want to assess whether there is a crosstalk between Snail1 and Prrx1, by which each factor promotes its own EMT mode, particularly by differential regulation of stemness.

Here, we describe a gene regulatory network (GRN) by which Snail1 directly represses *Prrx1* transcription, and Prrx1, through direct activation of the miR-15 family, attenuates Snail1 expression. We find that Snail1 is a direct target of these microRNAs (miRNAs) among different vertebrate species. miRNAs are short noncoding RNAs that posttranscriptionally regulate their target genes[13], and are crucial players in regulating cell plasticity and EMT[14]. We also find that this GRN triggers an expression switch from Snail1 to Prrx1, with Snail1 being an early response gene to EMT-inducing signals, followed by the activation of Prrx1 that in turn attenuates Snail1 expression. We support our findings by analyses in cultured cells, in vivo in different vertebrate embryos and public databases of cancer patients. We illustrate that this GRN rather than regulating the balance between epithelial and mesenchymal states as the previously described networks involving microRNAs, drives the selection of the EMT mode.

## Results

**Prrx1 and Snail1 are expressed in complementary patterns**. In zebrafish embryos, which bear two paralogs for each gene (*prrx1a* and *prrx1b; snail1a* and *snail1b)* due to the extra duplication in the teleost genome[3,15], we performed RNA in situ hybridization (ISH) and found a complementary expression pattern. In the developing somites where *snail* genes are abundantly expressed, *prrx1* genes expression are restricted to small cell populations where *snail* expression is low or absent (Fig. 1a). Although at 20-somite stage both *snail1* and *prrx1* are expressed in the cranial neural crest (Fig. 1a), transverse sections of double-fluorescent

ISH shows that they are also expressed in a complementary manner (Fig. 1b). Single-cell RNA sequencing (scRNA-seq) data from zebrafish embryos at 18 h post fertilization (hpf) (GEO: GSM3067194)[16] provides further evidence for this complementary expression of *snail1a/b* and *prrx1a/b* in the majority of cells, with a significant negative correlation (Fig. 1c, Supplementary Fig. 1a). This is compatible with our previous findings in the chicken embryo[8] (Fig. 1d).

To assess whether this complementarity was also conserved in mammals, we characterized the expression of Snail1 and Prrx1 in the mouse. At E8.5, embryos manifest a complementary expression pattern for *Snail1* and *Prrx1* at whole-mount level, particularly evident in the mesodermal populations along the medio-lateral axis and in the somites (Fig. 1e). Transverse sections of anterior regions (Fig. 1e) confirm this complementarity in the neural crest populations, where *Snail1* is expressed in the premigratory crest (arrow) and both are expressed in migratory neural crest populations (arrowheads) in a distinct manner. In more posterior tissues (Figs. 1e and dashed line2), the complementarity is evident in the mesodermal cells, where *Snail1* is highly expressed in the pre-somitic mesoderm (arrow) while *Prrx1* is expressed in the lateral plate mesoderm (LPM, arrowhead). Expression is also complementary at E9.5 mouse embryos in the somites and as in the chick, *Snail1* is highly expressed in immature somites where *Prrx1* expression is not evident. Also, LPM cells are positive for *Prrx1* expression but not *Snail1* (Supplementary Fig. 1b). Analysis of transverse sections at the level of the branchial arches, also showed complementarity, where in more internal cell populations *Snail1* was highly expressed concomitant with low *Prrx1* expression, while more ventral cell populations showed high expression of *Prrx1* and lower of *Snail1* (Supplementary Fig. 1c). A similarly complementary pattern was also evident in the mature somites. *Snail1* is highly expressed in the sclerotome (SC) and *Prrx1* in the dermomyotome (DM) (Supplementary Fig. 1d). The analysis of scRNA-seq from E9.5 embryos (GEO: GSE87038)[17] clearly shows this complementary expression at the single-cell level (Fig. 1f). These data confirm that *Snail1* and *Prrx1* are expressed in a conserved dynamic and complementary manner in cells undergoing EMT during vertebrate development.

To determine whether this complementary expression pattern holds true in pathology, we first looked at the expression levels of SNAIL1 and PRRX1 in human breast cancer cell lines[18], and did not find cells with both *SNAIL1* and *PRRX1* high. Supplementary Figure 1e shows examples of cell lines with high-SNAIL1 expression (SUM149PT); cells with intermediate/low levels of both (MDA436), and cells with high-PRRX1 expression (BT549). These three cell lines are derived from triple-negative/basal type breast tumors, and manifest mesenchymal properties[19], including migratory and invasive behaviors[8,20]. However, SUM149PT and MDA436 cells are metastatic and show cancer stem cell-like properties[21], while BT549 cells need to lose PRRX1 expression to acquire both stem cell properties and metastatic potential[8]. Thus, the comparison of the three cell lines for *SNAIL1* and *PRRX1* expression can provide insights into their potential regulation. Signal intensity analyses for SNAIL1 and PRRX1 proteins show a negative correlation in SUM149PT and BT549. Interestingly, MDA436 expresses both SNAIL1 and PRRX1 at similar intermediate levels (Supplementary Fig. 1f), altogether suggesting that SNAIL1 or PRRX1 cannot be co-expressed at high levels.

Next, we examined databases from breast cancer patients (GEO: GSE75688)[22], and found that *SNAIL1* and *PRRX1* depict a complementary expression pattern with a significant negative correlation (Fig. 1g). In addition, in a larger scRNA-seq dataset from head and neck carcinoma patients (GEO: GSE103322)[23], *SNAIL1* and *PRRX1* expressions were also complementary and

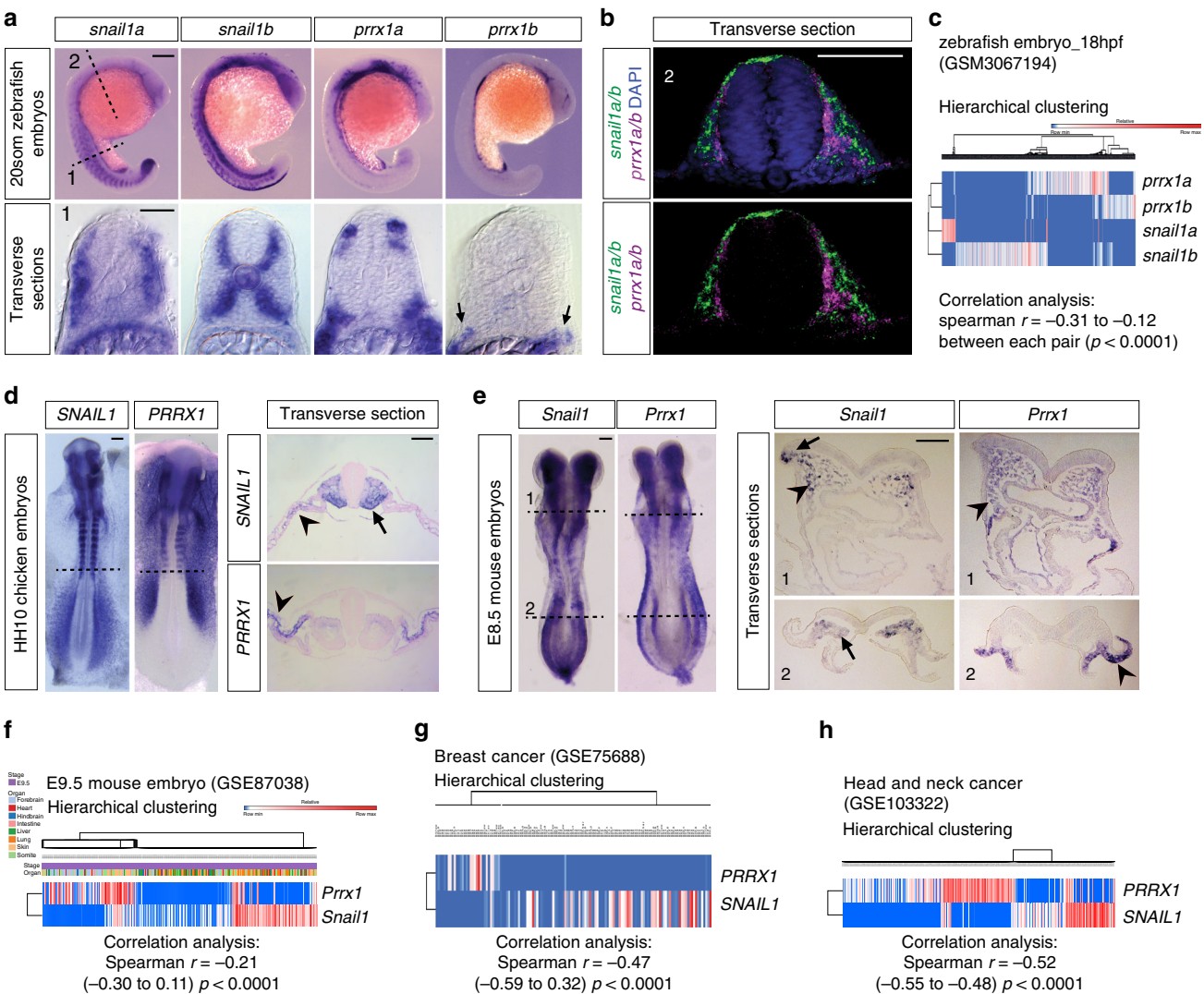

**Fig. 1** Snail1 and Prrx1 complementary expression in development and disease. **a** Lateral view of 20-somite zebrafish embryos showing *snail1a*, *snail1b*, *prrx1a* and *prrx1b* expression in whole-mount (top) and transverse sections (1), showing complementary patterns in somites. **b** Transverse section of a zebrafish embryo in the cranial neural crest region showing complementary expression of *snail1a/b* (green) and *prrx1a/b* (red) taken at the level indicated by (2) in (**a**) with or without DAPI staining (nuclei). **c** Heatmap showing hierarchical clustering of scRNA-seq data from 18 hpf zebrafish embryos, from public database GEO: GSM3067194, with significant negative correlations between gene pairs (detailed in Supplementary Fig. 1a). **d** Dorsal view of HH10 chicken embryos showing *PRRX1* and *SNAIL1* expression in whole-mount and transverse sections at the level indicated by dashed lines, showing complementary patterns for *SNAIL1* and *PRRX1* In the somites (arrow) and in the LPM (splanchnopleura and somatopleura, respectively; arrowheads). **e** Expression of *Snail1* and *Prrx1* in dorsal views of E8.5 mouse embryos. Transverse sections of E8.5 embryos from the regions indicated by dashed lines (anterior and posterior, 1 and 2, respectively), showing complementary expression of *Snail1* and *Prrx1* in premigratory (1, arrow), and migratory (1, arrowheads) neural crest (PNC and MNC, respectively) and mesodermal populations including presomitic mesoderm (2, arrow) and lateral plate mesoderm (2, arrowhead). **f** Hierarchical clustering heatmaps of scRNA-seq data from E9,5 mouse embryos for *Snail1* and *Prrx1* expression (GEO: GSE87038), showing a significant negative correlation. This dataset contains cells from different embryonic tissues, shown in different colors. **g, h** Heatmaps of hierarchical clusterings of single-cell RNA sequencing data from public datasets of breast (GEO: GSE75688) and head and neck carcinoma (GEO: GSE103322), showing mutually exclusive expression of Snail1 and Prrx1. The color scale shown in (**f**) is also valid for (**g**) and (**h**), representing the normalized values for the number of reads. Scale bars: 250 μm for whole mounts and 100 μm for sections. Statistical analyses for scRNA-seq data has been done using Spearman *r* correlation test, scRNA-seq single-cell RNA sequencing, vs. versus, hpf hours post fertilization. Source data are provided as a Source Data file

negatively correlated, with some small cell populations expressing both (Fig. 1h). Notably, in all the analyzed scRNA-seq datasets, extremely few cells were found to express both EMT-TFs at high levels (Supplementary Table 1).

Taken together, these findings suggest that there may be a regulatory mechanism between these two EMT-TFs that is conserved not only during vertebrate development but also in pathological EMTs. Therefore, we next wanted to understand the molecular basis of the complementary expression of Snail and

Prrx1. Considering that they are TF and there have been previous examples of mutually regulated TF pairs during embryonic development[24,25], we set up to study this possibility for Snail1 and Prrx1. To test this hypothesis, we first examined the response in terms of gene expression after acute downregulation of either SNAIL1 or PRRX1 in the same cell context. We used MDA436 cells, which express both TFs at low/intermediate levels, and found that SNAIL1 knockdown (KD) using short interfering RNA (siRNA) resulted in an increased expression of *PRRX1* and

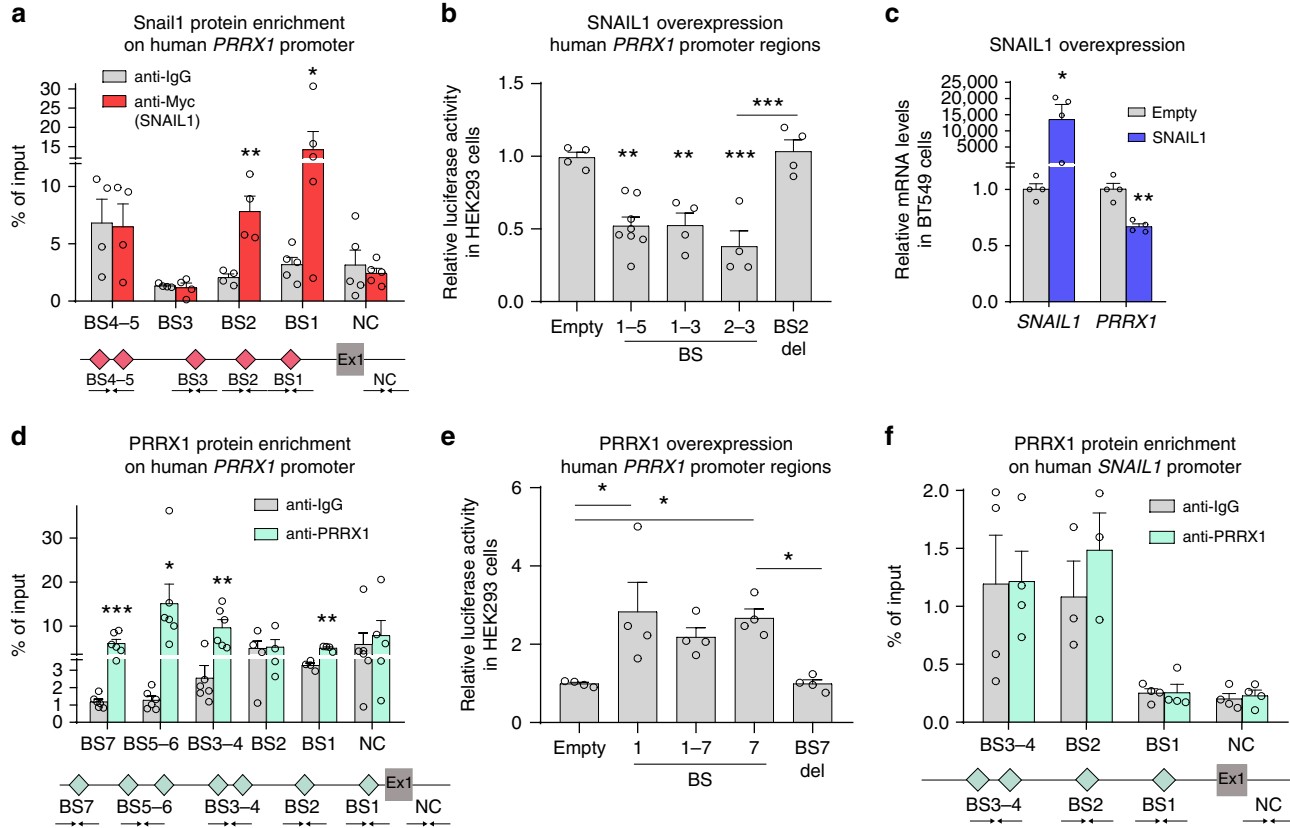

**Fig. 2** SNAIL1 and PRRX1 act antagonistically on *PRRX1* promoter. **a** Snail1 enrichment in the human *PRRX1* promoter shown by ChIP assay in BT549 cells, using anti Myc antibody (for Snail1-Myc overexpression). A schematic map is shown; red diamonds represent SNAIL1 potential binding sites (E-boxes; CANNTG) on *PRRX1* promoter. (BS1: −1689, BS2: −4102, BS3: −6753, BS4: −7277, BS5: −7318 and NC: +104560). Ex1: *PRRX1* exon 1 ($n = 4$). **b** Activity of different regions of the human *PRRX1* promoter after SNAIL1 transfection assessed by luciferase assays in HEK293 cells ($n = 4$, except for BS1–5 for which $n = 8$). **c** qPCR assay showing downregulation of *PRRX1* transcription upon SNAIL1 transfection in BT549 cells ($n = 4$). **d** PRRX1 directly binds to its own promoter, as assessed by ChIP assays in BT549 cells using a PRRX1 specific antibody. A schematic map is shown; cyan diamonds represent PRRX1 potential binding sites (TAATKDS) on its own promoter. (BS7: −6875, BS6: −5579, BS5: −5202, BS4: −4147, BS3: −4096, BS2: −1653, BS1: −1197 and NC: +104560). Ex1: *PRRX1* exon 1 ($n = 6$, except for BS1–2 for which $n = 4$). **e** Activity of different regions of the human *PRRX1* promoter after PRRX1 overexpression in luciferase assays in HEK293 cells ($n = 4$). **f** Lack of enrichment for PRRX1 binding to human *SNAIL1* promoter assessed by ChIP assays in BT549 cells (BS4–1: −1753, −1621, −954, −198, and NC: +11799). Ex1: *SNAIL1* exon 1 ($n = 4$). Locations of red and cyan diamonds represent distances between BSs and the promoter. Arrows represent primers used for qPCR amplification. BS biding site, NC negative control region, which does not contain potential BSs. Symbols in binding sites are as follows. K: T/G, D: G/A/T, S: G/C and N: G/A/T/C. Bars represent mean plus standard error of the mean (SEM), indicated (n) represent number of independent experiments as biological replicates and asterisks indicate significant *p* value in *t* test compared to the control in each test for (**a**), (**e**), (**d**), (**f**) and ANOVA with Bonferroni's multiple comparison test for (**b**), (**e**) (**p* < 0.05, ***p* < 0.01 and ****p* < 0.001). Source data are provided as a Source Data file

similarly, PRRX1 KD resulted in *SNAIL1* upregulation (Supplementary Fig. 1g). These observations prompted us to investigate the molecular relationship between Snail1 and Prrx1.

**Snail1 directly represses *Prrx1*.** As Snail1 has been described as a potent transcriptional repressor[26], we examined whether it could directly target *Prrx1* transcription. We looked for consensus Snail1 E-boxes[27] within the mouse and human *Prrx1* promoters and putative enhancer regions, and found several predicted binding sites (BS). To narrow down the number of BSs we used ENCODE data from different cell contexts[28] using UCSC genome browser (https://genome-euro.ucsc.edu/index.html), and focused on the regions that were conserved and positive for DNase-I hypersensitivity and H3K27Ac marks, suggesting an active chromatin region. We performed chromatin immunoprecipitation (ChIP) assay in NIH3T3 mouse fibroblast cells transiently overexpressing Snail1, and found several regions with significant enrichment for Snail1 protein occupancy in the mouse *Prrx1* promoter; BS1–3 (Supplementary Fig. 2a), which results in

repression of promoter activity showed using dual luciferase reporter assay (Supplementary Fig. 2b). We then analyzed the human *PRRX1* promoter and performed ChIP in BT549 cells transiently overexpressing Snail1, and found at least two sites with significant enrichment for its protein occupancy, BS1–2 (Fig. 2a). Snail1 binding leads to repression of the promoter activity, which is abolished upon deletion of BS2 (Fig. 2b). In agreement with this, *PRRX1* was downregulated after SNAIL1 transfection in BT549 (Fig. 2c). Altogether, these results indicate that Snail1 directly binds to, and represses *Prrx1* promoter activity both in mouse and human cells.

**Prrx1 induces its own expression.** When we searched for Snail1 E-boxes in both mouse and human *Prrx1* promoters, we also found several predicted Prrx1 BSs (TAATKDS)[29] (Fig. 2d and Supplementary Fig. 2c). Thus, we next examined whether Prrx1 could regulate its own transcription. We used above-mentioned strategy to narrow down the potential BSs. We performed ChIP for endogenous chromatin bound Prrx1 protein in NIH3T3, and

found two regions with enrichment for Prrx1 occupancy in its own promoter (BS3–4, Supplementary Fig. 2c). In human BT549 cells, endogenous PRRX1 protein occupancy was significantly enriched in several BS (Fig. 2d). Luciferase assay for both mouse and human sequences show an activation of the *Prrx1* promoter upon Prrx1 overexpression (Fig. 2e and Supplementary Fig. 2d) and this activation no longer occurs when BS7 is deleted (Fig. 2e).

Next, we wanted to know whether there was a reciprocal regulation between Snail1 and Prrx1, and we examined Prrx1 protein binding in the *Snail1* promoter. Although we found putative Prrx1 BSs within the *Snail1* proximal promoter that met our criteria, ChIP in both mouse (Supplementary Fig. 2e) and human cells (Fig. 2f) failed to show Prrx1 protein binding enrichment in those sites. In addition, activity assay for *Snail1* promoter shows no significant change when Prrx1 is over-expressed (Supplementary Fig. 2f). Altogether, these results indicate that Prrx1 is able to bind to its own promoter and enhance its own expression, but it does not bind to *Snail1* promoter, suggesting that Prrx1 is not a direct *Snail1* transcriptional repressor. In fact, Prrx1 has been described as a transcriptional activator[30], which suggests a putative indirect regulation of Snail1 expression by Prrx1.

**Prrx1 directly induces expression of *miR-15* family members.** To investigate downstream targets of PRRX1, we used MDA-MB-231 breast cancer cells which express low level of PRRX1 to generate a stable cell line where PRRX1 was ectopically over-expressed (MDA231-PRRX1), and performed comparative microarray analyses (GEO: GSE138078). *SNAIL1* was down-regulated after PRRX1 overexpression, reinforcing the idea of a mutual negative regulation. Interestingly, we found many miR-NAs that were upregulated in MDA231-PRRX1 cells (Fig. 3a and Supplementary Data 1). As miRNA are potent regulators of gene expression[13], we hypothesized that Prrx1 may repress Snail1 via recruiting miRNAs. We selected for further analyses miR-424 and miR-503 that are members of miR-15 family (hereafter referred as Mir-15-P1d and Mir-15-P2d, as recently suggested[31]; Supplementary Table 2) because they were predicted to target *Snail1*, are conserved in different vertebrates, have a relatively well known promoter, and most importantly because their studied functions as regulators of stemness, invasion, migration and cell proliferation[32–34], are compatible with those described for PRRX1[8]. We validated the microarray data by qPCR analyses (Fig. 3b).

To test whether these miRNAs could be directly induced by PRRX1, we examined the expression of their precursor miRNAs (Pre-miRNAs) upon transient overexpression of mouse Prrx1 in MDA231 cells. To do so, we generated an inducible lentiviral system (Tet-ON) including P2A peptide and overexpressed both Prrx1 and nuclear yellow fluorescent protein (nYFP)-P2A-Prrx1 (MDA231 conditional Prrx1; MDA231cP) upon doxycycline (dox) treatment. Forty-eight hour after dox induction, YFP-positive cells were FACS sorted and subjected to qPCR, showing that, as expected, Prrx1 was overexpressed and also that *Pre-miRNAs* were significantly upregulated (Fig. 3c and Supplementary Fig. 3a), suggesting that they may be direct Prrx1 targets.

As miRNA molecules of the same family share the seed sequence (nucleotides 2–7) and they can potentially target the same mRNAs, we examined whether the expression of other miR-15 family (miR-15f) members (Supplementary Fig. 3b) also correlated with that of PRRX1. Thus, we performed transient KD of PRRX1 in BT549 cells with a specific short-hairpin RNA[8] (shRNA) vector containing nYFP. We then FACS-sorted positive cells, and checked pre-miRNAs expression for an early transcriptional response. We observed that *PRRX1* transcript was significantly reduced, and also that all the members of the miR-

15f were significantly downregulated upon PRRX1 KD (Fig. 3d). We also confirmed that this was the case in HEK293 cells (Supplementary Fig. 3c).

To assess whether PRRX1 directly binds the *miR-15f* promoters, we performed ChIP assays in BT459 on the human *Mir-15-P1/2d* promoter. We found enrichment for PRRX1 binding in at least one putative BS, indicating that PRRX1 can directly bind to the *Mir-15-P1/2d* promoter (Fig. 3e). Also, PRRX1 overexpression increased the activity of the *Mir-15-P1/2d* promoter, and the deletion of PRRX1 potential BS prevented this activation (Fig. 3f). On the other hand, KD of PRRX1 in BT549 cells showed a reduction in promoter activity, confirming that PRRX1 enhances *Mir-15-P1/2d* transcription (Supplementary Fig. 3d). We also performed ChIP and Luciferase assays in mouse NIH-3T3 cells and confirmed that this direct binding and activation of the *Mir-15-P1/2d* promoter is conserved between human and mouse (Supplementary Fig. 3e, f). Furthermore, ChIP assays for Prrx1 in the promoter regions of other members of the miR-15f in human and mouse, show that Prrx1 can directly bind and likely activate other members of the family (Supplementary Fig. 3g–i). Altogether, these experiments indicate that PRRX1 is able to directly bind to the promoter, and activate the transcription of *miR-15f* members.

To elucidate in vivo the relationship between Prrx1 and miR-15f members we studied the expression of the miRNAs by ISH using specific DIG-labeled LNA (locked nucleic acid) probes to detect their mature versions in mouse embryos. At E8.5, expression can be detected in similar territories to those of *Prrx1*, e.g., MNC cells (Fig. 3g), suggesting that Prrx1 induces the expression of these miRNAs in those territories. Altogether these results indicate that miR-15f members are directly induced by Prrx1 in vitro and in vivo.

**Prrx1 indirectly attenuates *Snail1* expression through *miR-15*.** We validated our microarray data (Fig. 3a) by qPCR, and confirmed that in MDA231-PRRX1cells, *SNAIL1* is significantly downregulated (Supplementary Fig. 4a). To assess whether this was an early response, we examined *SNAIL1* expression in the inducible Prrx1 system. We analyzed MDA231-cPrrx1 cells after 48hs of dox induction and found that *SNAIL1* was downregulated, confirming the microarray data also in a short-term experiment (Fig. 4a).

The observed downregulation of *SNAIL1* after constitutive and short-term induction of PRRX1 expression together with the fact that Prrx1 directly induces the transcription of *miR-15f* members that were predicted to target *Snail1* (Supplementary Fig. 4b), suggested that Prrx1 could be attenuating Snail1 expression through the activation of miR-15f members. Importantly, the miRNA responsive element (MRE) in *Snail1* 3′ untranslated region (UTR) is conserved in all vertebrates (Supplementary Fig. 4c), and the in silico prediction for binding of human, mouse, chicken, and zebrafish miR-15f members to the corresponding *Snail1* 3′ UTRs suggest a strong hybridization not only in the seed region but also in the 3′ end of the miRNAs (Supplementary Fig. 4b, d–f). Thus, we examined whether *Snail1* could be directly targeted by miR-15f. We first overexpressed *Mir-15-P1/2d* in MDA231, and found a significant downregulation of *SNAIL1*. As a control, we overexpressed a seed-mutated version of the miRNAs (*Mir-15-P1/2d*-mut), unable to bind to *SNAIL1* 3′ UTR (Fig. 4b). Conversely, we used specific miRNA inhibitors/LNAs to KD *Mir-15-P1/2d* in BT549, which express high levels of PRRX1 and miR-15, and that resulted in upregulation of *SNAIL1* (Fig. 4c). To confirm that this downregulation was through direct binding of the miRNAs to the 3′UTR, we performed Luciferase assays and observed that overexpression of *Mir-15-P1/*

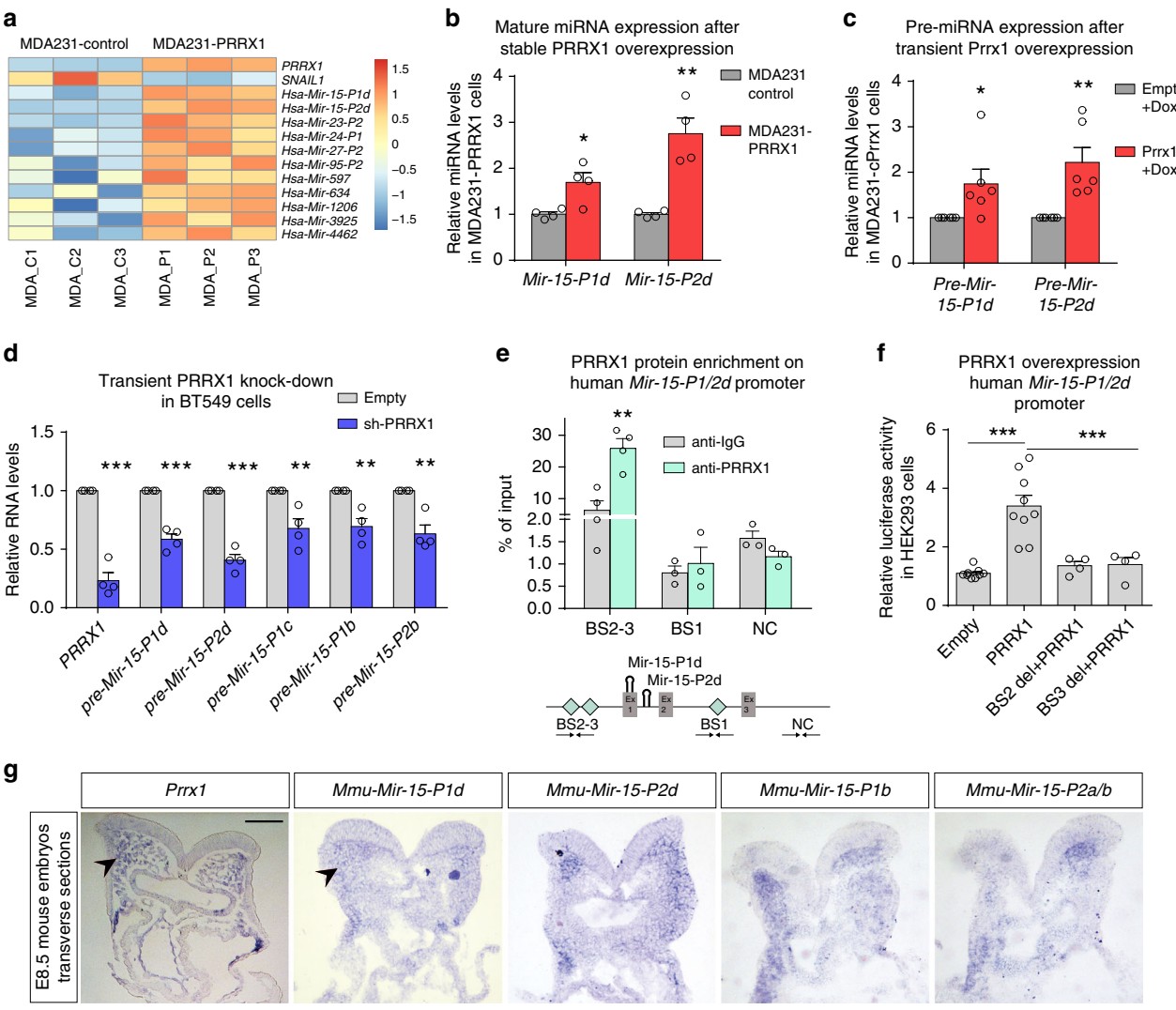

**Fig. 3** Prrx1 directly induces the expression of *miR-15* family. **a** Heatmap showing Robust Multi-array Average (RMA) normalized probe intensity values of *PRRX1*, *SNAIL1*, and selected miRNAs in MDA231 cells, with three control samples (MDA_C1 to C3) and three samples in which PRRX1 was overexpressed (MDA_P1 to P3). The intensities mapped as color scale show normalized fold change with respect to the average ($n = 3$). **b** Validation by Taq-Man qPCR of *Mir-15-P1d* and *Mir-15-P2d* upregulation after PRRX1 stable overexpression in MDA231 cells lines, using specific probes to detect the mature miRNAs ($n = 4$). **c** *Mir-15-P1d_pre* and *Mir-15-P2d_pre* upregulation upon conditional (Dox-mediated) Prrx1 overexpression in MDA231 cells ($n = 6$). **d** qPCR assay showing downregulation of *PRRX1* and *premiRNAs* transcription upon transient knockdown (KD) of PRRX1 in BT549 cells. *PRRX1* shRNA plus YFP transfected cells were sorted after 4 days ($n = 4$). **e** PRRX1 directly binds to the human *Mir-15-P1/2d* promoter as shown by ChIP assay in BT549 cells using a PRRX1 specific antibody. Cyan diamonds in the schematic map represent distances between PRRX1 potential BSs and the promoter potential motifs, TAATKDS, on *Mir-15-P1/2d* promoter. (BS1: −293, BS2: −949, BS3: −1451 and NC: +1893). Ex1–3 represent exons of the host long noncoding RNA *MIR503HG*. Arrows represent primer sets used for ChIP detection. ($n = 3$ except for BS2–3 for which $n = 4$). **f** Activation of human *Mir-15-P1/2d* promoter by PRRX1 overexpression shown by dual luciferase assay in HEK293 cells. This activation is abolished upon deletion of the PRRX1 binding sites in *Mir-15-P1/2d* promoter (del1/2 + PRRX1) ($n = 4$ except for deletions for which $n = 4$). **g** Transverse sections of the cranial region of E8.5 embryos showing the expression of *Prrx1*, *Mir-15-P1d*, *Mir-15-P2d*, *Mir-15-P1b*, and *Mir-15-P2a/b* in similar regions (arrowheads). Scale bar: 100 μm. Dox doxycycline, sh short hairpin RNA (shRNA), BS binding site, del deletion. Bars represent mean plus SEM, indicated ($n$) represent number of independent experiments as biological replicates and asterisks indicate significant $p$ value in $t$ test for (**b**–**e**) and one-way ANOVA with Bonferroni's multiple comparison test for (**f**). (*$p < 0.05$, **$p < 0.01$ and ***$p < 0.001$). Source data are provided as a Source Data file

*2d* repressed human *SNAIL1* 3′ UTR activity. This repression was abolished when we used either the seed-mutant miRNAs or a MRE-mutated version of the *SNAIL1* 3′ UTR (Fig. 4d). Moreover, we checked other miR-15f members and found that *Mir-15-P1/2b* behaves similarly (Supplementary Fig. 4g). This regulation is conserved in mouse, chicken and zebrafish, as overexpression of *miR-15f* members can repress *Snail1* 3′ UTR from those species (Supplementary Fig. 4h–j), indicating that *Snail1* mRNA is a bona fide target of miR-15f in vertebrates.

To test whether this regulation operates in vivo, we knocked-down *miR-15f* in zebrafish embryos by injecting a sponge RNA[35] containing partially complementary sequences to all *miR-15f* members (Supplementary Fig. 4k). Zebrafish embryos were injected at 1–2 cell stage and then collected at 20-somite stage. qPCR from pools of embryos confirmed that *snail1a/b* transcripts were more abundant in those injected with sponge (Fig. 4e). ISH analysis showed that in one third (30/96) of the injected embryos, *snail1a/b* expression was increased (Fig. 4f). Transverse sections

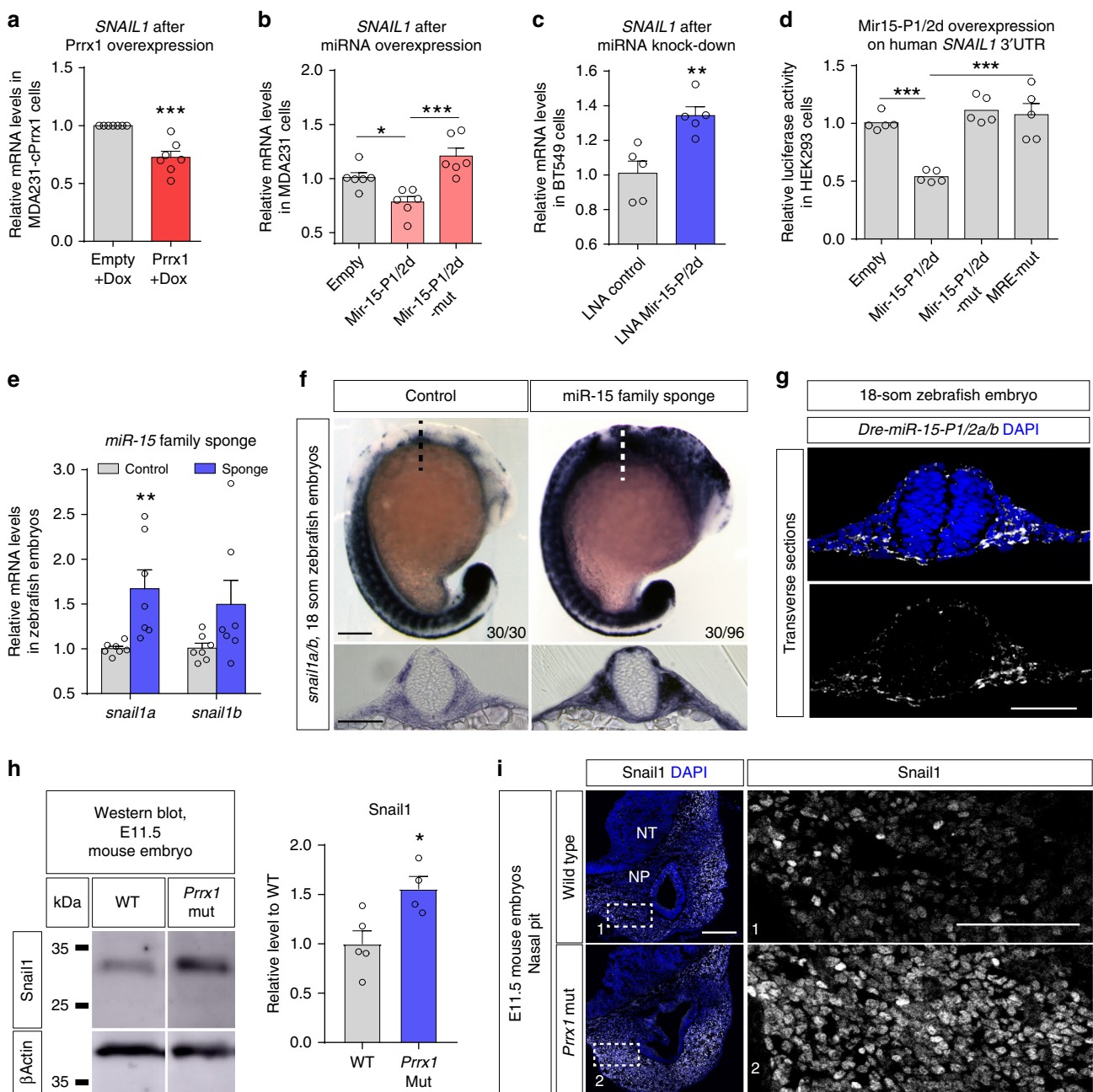

Fig. 4 Prrx1 attenuates *Snail1* expression through the activation of *miR-15* family members. **a** qPCR assay showing the downregulation of *SNAIL1* upon conditional overexpression of Prrx1 in MDA231, induced by doxycycline after 48 h (*n* = 7). **b** qPCR assay showing downregulation of *SNAIL1* upon transient overexpression of *Mir-15-P1/2d* in MDA231 cells, using seed mutated miRNAs as control (*Mir-15-P1/2d-mut*) (*n* = 6). **c** qPCR assay showing upregulation of *SNAIL1* upon transient KD of in *Mir-15-P1/2d* in BT549 cells using miRNA LNA inhibitors (*n* = 5). **d** Repression of human *SNAIL1* 3′ UTR after *Mir-15-P1/2d* overexpression, using seed mutated miRNAs (*Mir-15-P1/2d-mut*) or MRE mutant *SNAIL1* 3-UTR (*MRE-mut*) as control (*n* = 5). **e** qPCR analysis shows upregulation of *snail1a/b* expression in the sponge-injected (*miR-15* family knocked-down) zebrafish embryos compared to controls (*n* = 6). **f** 20-somite control or miR-15 family sponge-injected zebrafish embryos showing combined *snail1a/b* expression by in situ hybridization. Transverse sections taken at the levels indicated by the dashed lines. Scale bar: 250 μm. **g** Fluorescent miRNA in situ hybridization to detect mature miRNAs in transverse sections of 20-somite zebrafish embryo, showing the expression of *Dre-miR-15-P1/2a/b* in neural crest derived cell populations. Scale bar: 250 μm. **h** Western blot for Snail1 and β-actin from E11.5 control and *Prrx1* mutant embryos, including quantification of intensity of Snail1 signal (*n* = 5 WT, 4 KO). **i** Snail1 IF in the nasal pit regions of E11.5 WT and mutant embryos. Scale bar: 250 μm for NP sections and 100 μm for insets (boxes 1 and 2). MRE miRNA responsive element, LNA locked nucleic acid, WT wild type, Mut mutant, NT neural tube, NP nasal pit. Quantifications are performed for one section of WT or mutant embryos, and the increase and expansion is observed in *n* = 2/2 mutant embryos compared to *n* = 3/3 different E11.5 WT. Bars represent mean plus SEM, indicated (*n*) represent number of independent experiments as biological replicates and asterisks indicate significant *p* value in *t* test for a, c, e, h and one-way ANOVA with Bonferroni's multiple comparison test for (**b**), (**d**) (**p* < 0.05, ***p* < 0.01 and ****p* < 0.001). Source data are provided as a Source Data file

confirmed the drastic increase in *snail1* expression in the territories of endogenous *miR-15* expression (Fig. 4f, g). Some embryos showed ectopic *snail1* positive cells that migrated beyond their normal position (Supplementary Fig. 4l, asterisk), as may be expected for a Snail gain of function. These findings are compatible with *miR-15f* attenuating *snail1* expression in zebrafish, confirming the results previously observed in cultured cells and living embryos.

To assess whether the loss of Prrx1 protein also affects Snail1 levels in the mouse, we performed western blot analysis in *Prrx1* mutant embryos (bearing a LacZ knock-in in the *Prrx1* locus (*Prrx1tm1Jfm*)[36]) (Supplementary Fig. 5a), and found a significant increase of Snail1 protein levels compared with WT embryos (Fig. 4h and Supplementary Fig. 9). Snail1 IF in cryo-sections from E11.5 embryos (Fig. 4i and Supplementary Fig. 5b, d) showed an increased expression and expansion of Snail1 territories in the head of mutant embryos, compatible with a mutual Prrx1/Snail1 repression (Supplementary Fig. 5b–e and Fig. 4i). Altogether, these findings are compatible with our in vitro analyses and indicate that Prrx1 indirectly attenuates the expression of Snail1 through the induction of miR-15f, a mechanism that is conserved in vertebrates.

### SNAIL1 and PRRX1-miR-15 correlation with patients' prognosis.
As shown previously in breast and in lung squamous cell carcinoma patients, high expression of *SNAIL1* in tumors correlate with poor prognosis[8,11], while high *PRRX1* correlates with good prognosis[8]. We confirmed this notion analyzing breast cancer patients' overall survival (OS) from a different dataset[37] containing different clinical conditions and subtypes, including tumoral grade, ER/PR/HER2, lymph-node status, etc. (Total) (Fig. 5a). These correlations are even stronger when we analyze patients with lymph-node positive status (Fig. 5b). We also found that in terms of patients' OS the expression of *miR-15f* members have similar profile to that of *PRRX1*, i.e., high expression of *miR-15f* members correlate with better OS in total and lymph-node positive patients' groups (Fig. 5a, b).

Breast cancer is highly heterogeneous, including several different subtypes. Among all subtypes, triple-negative breast cancer (TNBC) is classified as basal-like[38] and has been associated with EMT features and mesenchymal phenotype, also manifesting worst prognosis and lowest response to therapy[39,40]. Therefore, we examined the expression of *SNAIL1/PRRX1* and *miR-15f* members in patients with basal/TNBC subtype. Similar to the results obtained with total patients, high expression of *SNAIL1* significantly correlated with poor OS, while *PRRX1* high expression was concomitant with more favorable OS (Supplementary Fig. 6). Also, three members of the *miR-15f*, out of the four that were shown in total patients' group, followed *PRRX1* expression trend (Supplementary Fig. 6). Altogether, these results indicate that similar to *PRRX1*, when *miR-15* are expressed at high levels in cancer patients, they are associated with better prognosis, even in patients who have the cancer spread to their lymph nodes and also in basal/TNBC subtype which comprises the most unfavorable disease.

### Snail1 and Prrx1 are sequentially expressed during EMT.
Snail1 seems to be the first EMT-TF to be expressed in regions of the embryo that will undergo an EMT process, including the primitive streak and the neural crest[41]. *Snail* genes (*Snail1* in the mouse and *Snail2* in the chick) are expressed in the PNC before delamination from the neural tube and continue to be expressed in early migratory cells but their expression decays along the migratory routes[41,42]. In contrast, Prrx1 is expressed in migratory crest subpopulations (MNC; ref. [8] and this work). The expression

of these two EMT-TFs follows a temporal order in the migration of the neural crest, and thus, likely reflects a hierarchy of activation during the EMT process that drives neural crest delamination and migration. To test this idea, and taking into account that both Snail and Prrx are activated by the TGF-β superfamily members bone morphogenetic proteins (BMPs) during embryonic development[8,12], we first treated developing zebrafish embryos from 4-somite stage with BMP and examined the levels of *snail1*, *prrx1* and *miR-15f* members by qPCR. *snail1b* was quickly upregulated, while *prrx1a/b* and *miRNAs* were induced later, concomitant with downregulation of *snail* genes (Fig. 6a).

To test this hypothesis in other species, we implanted BMP-coated beads close to somites in developing chicken embryos, and monitored the expression of these two EMT-TFs at different time points. *SNAIL1* was quickly upregulated 1 h after bead implantation, whereas *PRRX1* upregulation was observed after 5 h (Fig. 6b), supporting the notion of sequential activation of these EMT-TFs. In line with our findings in zebrafish, *SNAIL1* transcripts were reduced in the somites close to the bead after longer BMP treatment, while *PRRX1* was still upregulated in similar somite regions (Fig. 6b). We also found that the peak of SNAIL1 protein expression occurred after 5 h, and that of PRRX1 protein after 10 h (Supplementary Fig. 7a, b). This suggests that although *SNAIL1* transcription is quickly upregulated, the increase in protein level takes longer, allowing a slower *PRRX1* activation by BMP, that will gradually outcompete SNAIL1 expression.

To examine whether this sequential activation was also conserved in the mouse, we used a transgenic model containing a *downstream enhancer* of *Snail1* (DES) that drives the expression of enhanced green fluorescent protein (EGFP) to its endogenous territories, recapitulating the vast majority of *Snail1* expression in the developing embryo (Snail1 (DES)-EGFP reporter mouse; Supplementary Fig. 8). The half-life of the GFP protein is very long[43] whereas Snail1 protein is very unstable[44], thus, this model can provide a short-time lineage tracing system. In transverse sections of E8.5 embryos, we observed that PNC and early MNC cells expressed Snail1 and GFP, while late MNC cells are negative for Snail1 protein but positive for both GFP and Prrx1 (Fig. 6c). This observation is compatible with the idea that Snail1 and Prrx1 are expressed in a sequential manner, as Prrx1 positive cells seem to be descendants of Snail positive cells (double-labeled cells in Figs. 6c box4). Altogether, these results suggest that during embryonic development, cells undergo EMT by activating Snail1 first, and then attenuation of Snail1 is indirectly mediated by Prrx1 though the activation of the miR-15f, all being part of a GRN in which miR-15f may coordinate the transition from Snail1- to Prrx1-mediated EMT programs (Fig. 6d).

### Discussion
Epithelial cells transition to a variety of mesenchymal states to fulfill different roles in different contexts. The existence of diverse mesenchymal cells allows for a high degree of cell heterogeneity in terms of potential fates and therefore, functions. They are all called mesenchymal based on morphology, the expression of particular markers (although there is no universal marker to define them), and very often, their ability to migrate. Differences arise from the tissue of origin as these phenotypic transitions are governed by extracellular signals that activate a plethora of EMT-TFs.

Here, we show that Snail1 and Prrx1 are expressed in a complementary manner during vertebrate development and in cancer patients. Considering that all EMT-TFs are activated by the same extracellular signals, this strongly suggests that Snail1 and Prrx1 may be part of a mutual regulatory network. In this work, we describe such a mechanism. We find that Snail1 and Prrx1 behave

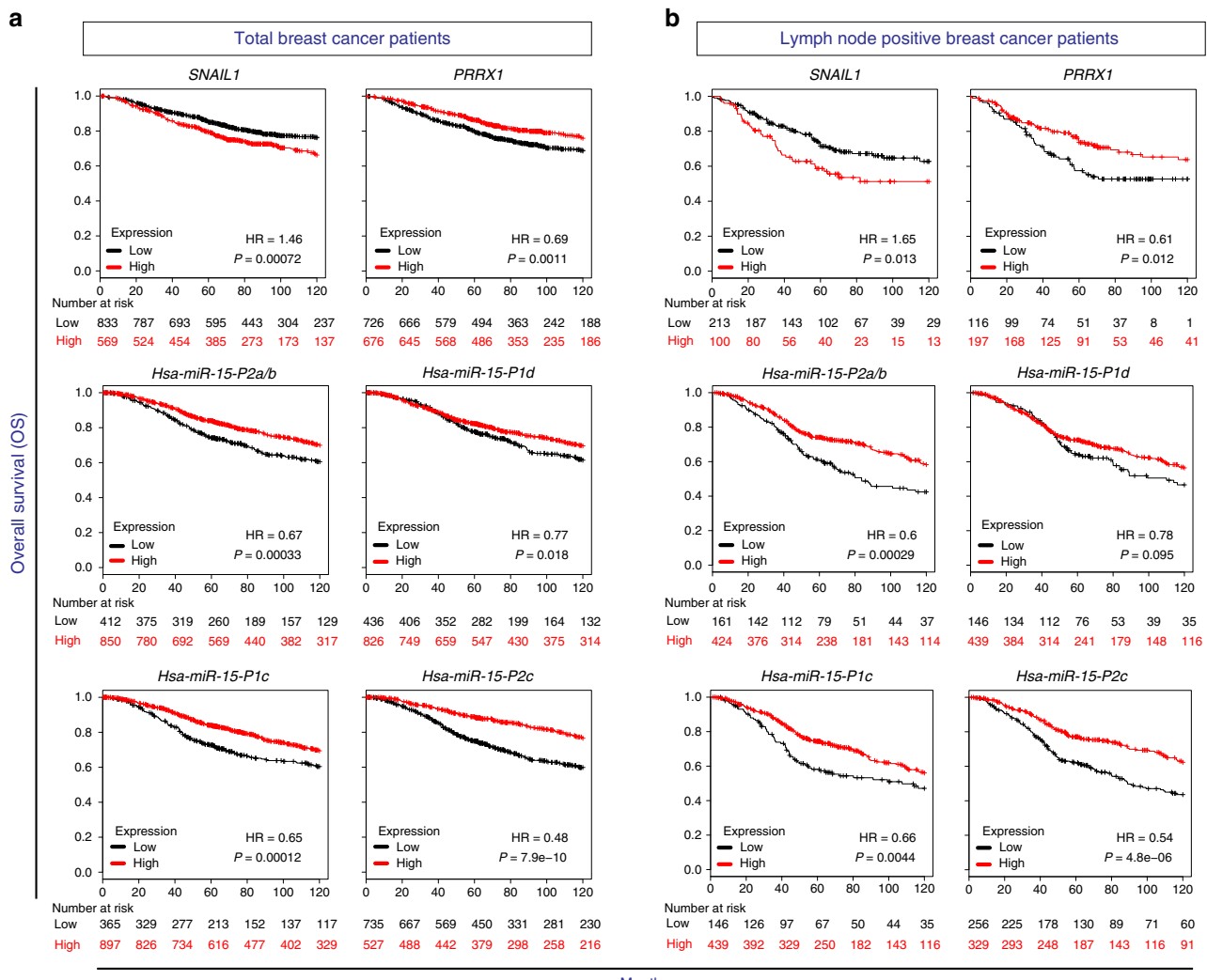

**Fig. 5** Relationship between the expression of *SNAIL1*, *PRRX1*, and *miR-15* family members and overall survival in breast cancer patients. Kaplan–Meier overall survival (OS) plots from breast cancer patients with lymph-node positive status, showing that high expression of *SNAIL1* correlates with low survival, while *PRRX1* high expression correlates with a better survival. The expression of *miR-15* family members follow a similar trend as that of *PRRX1*. Hazard ratio (HR) and logarithmic ranked *p* Value (longrank *P*) were analyzed to infer the significance of the differences. Numbers below each graph represent number of patients at risk in any given time (months), black for low expression and red for high expression of each gene/miRNA. The cut-off is automatically calculated based on the best performing threshold

as mutual repressors, albeit they use different mechanisms. Although in some contexts Snail1 can function as an activator[45], it is mostly a strong transcriptional repressor[26,46], and this is the mechanism that it uses to repress *Prrx1* expression. Prrx1, in turn, is a transcriptional activator[30] and accordingly, here we show that it represses Snail1 indirectly through the activation of a repressor, miR-15f. As miR-15f is composed of eight members in four different clusters, this provides a certain degree of variability in the repression, e.g., one or several family members can be used in specific cell contexts to exert a particular level of repression.

Following the developmental time in a particular tissue, it is possible to infer potential regulatory scenarios in which the mutual repression of these TFs could be integrated. For instance, in the mouse, Snail1 is expressed in the PNC and Prrx1 is expressed in subpopulations of MNC. During somite formation, Snail1 is expressed in the precursors of the whole somite and, later on in the mature somites, it is expressed in the SC and Prrx1 in the DM, also showing complementary expression. This implies that Prrx1 is activated in some descendants of Snail1 expressing cells. Altogether, this points to the successive upregulation of

EMT-TFs during the development of the neural crest and the somites, providing a hierarchical temporal order in the activation of these two TFs. As such, we provide evidence of this temporal hierarchy both in vitro and in vivo, with Snail1 being activated at earlier time points in response to an EMT-inducing signal.

Interestingly, the inducing signal is the same for Snail1 and Prrx1 (the TGF-β superfamily[8,12] and this work), and thus, the question is why they are not simultaneously activated when the signal is available in the embryo or in cancer cells. One explanation is that Snail1 bears a poised promoter[47], explaining why it usually is an early response gene activated shortly after TGF-β administration[48]. Snail1 fast activation may set an inhibitory scenario for Prrx1 upregulation, which may change when the duration of the TGF-β signal accumulates sufficient stimulus to induce Prrx1. This establishes a temporal order of predominant EMT programs, firstly mediated by Snail1, a strong epithelial repressor needed for cells to detach from their neighbors. Subsequently, Prrx1, a strong mesenchymal inducer, is activated and in turn, it will promote the acquisition and maintenance of robust mesenchymal features for cells to migrate to their destination.

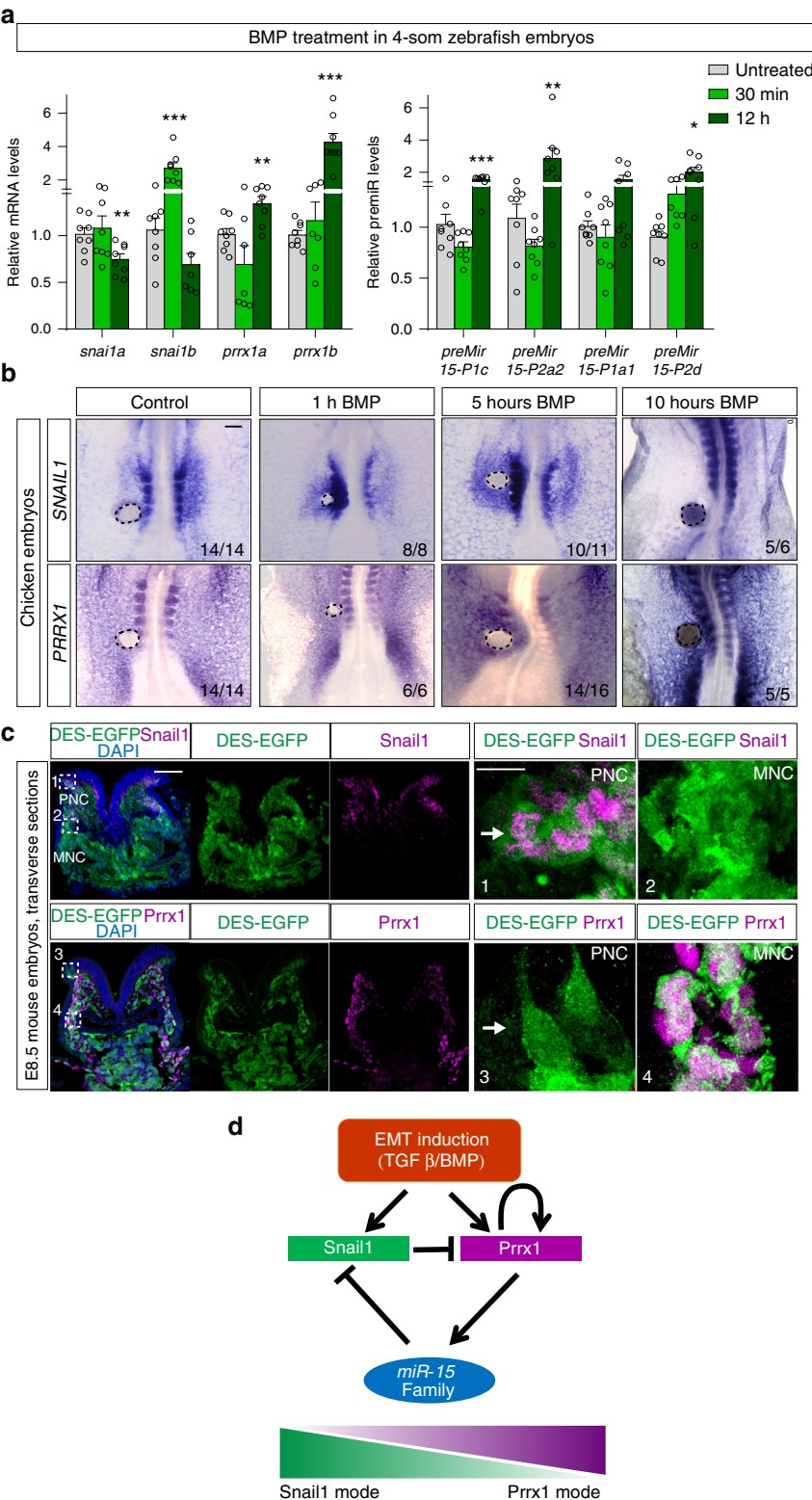

**Fig. 6** Snail1 and Prrx1 are sequentially expressed during EMT. **a** qPCR assay showing the expression of *prrx1a/b* and *snail1a/b* (left) and *premiRNAs* (right) in zebrafish embryos after different treatments with BMP (*n* = 8). Asterisks indicate significant *p* value in one-way ANOVA with Bonferroni's multiple comparison test compared to the control. (**p* < 0.05, ***p* < 0.01 and ****p* < 0.001). Source data are provided as a Source Data file. **b** Dorsal view of HH9 chicken embryos showing *SNAIL1* and *PRRX1* expression in control embryos and at different times after BMP-soaked bead implantation. **c** Transverse sections of E8.5 DES-EGFP reporter mouse embryos showing IFs for GFP and Snail1 (upper panel) or Prrx1 (lower panel). Insets show higher magnification pictures for GFP and Snail1 (1 and 2), or for GFP and Prrx1 (3 and 4). **d** Schematic model of the gene regulatory network. DES downstream enhancer of Snail1, PNC premigratory neural crest shown by arrows, MNC migratory neural crest. Scale bars: 100 μm. Scale bars: 100 μm for sections and 10 μm for insets (boxes 3–4)

Cells do not only express these two TFs but also members of the other EMT-TF families in a context-dependent combination that will determine their final phenotype. Nonetheless, Snail1 and Prrx1, as shown here, are mutually exclusive, thereby promoting their own mode of EMT. We also describe in this work that Prrx1 further reinforces its own program by activating its own promoter. Interestingly, Snail1 represses its promoter, thereby modulating its own expression[49]. This can also contribute to the establishment of the temporal hierarchy of EMT-TF activation, promoting Prrx1 upregulation, as we observe during embryonic development. Altogether, these mutual regulations and the response to the inducing signal establish a GRN that controls the EMT programs.

Snail1 and Prrx1 EMT programs differ in some important aspects. As such, EMT has been associated with the acquisition of stem cell properties[6] but Prrx1 represses stemness[8,9]. This impinges into the plasticity required in both embryonic and cancer cells to revert to a more epithelial phenotype at the site of their final destination either to differentiate into different organs during development or to form macrometastases during cancer progression (see refs. 8 and 50–52). Prrx1 needs to be downregulated in cancer cells to promote the reversion, and its downregulation implies an increase in stem cell properties that are also essential for metastatic growth. This explains why there is a positive correlation between high levels of Prrx1 and better prognosis in cancer patients[8]. Those cells may be stuck at a very mesenchymal state, lacking the plasticity required to colonize and restart proliferation in distant organs. In addition, as we have found that miR-15f members are direct targets of Prrx1, Prrx1 high levels will lead to miR-15 activation, resulting in Snail1 attenuation, promoting a decrease in stem cell properties and therefore in metastatic abilities. In agreement with that, we find that miR-15 expression also correlates with better survival rate in patients. However, we cannot exclude that other regulatory circuits impinging on Snail1 expression also play a role, including the well described Snail1/miR-34 and miR-200 reciprocal feedback loops[14].

Importantly, the relationship between EMT and stemness has been mainly studied in adult mammary gland and in cancer cells[6], but these differences in conferring stem-like cell properties may also have a role during embryonic development. Snail1 expressing cells might be in more stem state, as they can give rise to cells with a variety of fates. For instance, Snail1 is expressed early in primitive streak and the premigratory neural crest, both of which later give rise to a large variety of cell types[53]. As such, it has been proposed that the premigratory neural crest may still retain characteristics of the pluripotent blastula stage[54]. These multipotent territories do not express Prrx1, which is rather expressed in cells that have been already committed to particular lineages such as specific mesodermal or crest subpopulations. This is also compatible with the already discussed temporal hierarchy in the activation of these two EMT-TFs that occurs in parallel with commitment to different fates.

As much as Snail1 and Prrx1 trigger different EMT programs[8], they have important common functional roles inducing the transition towards a mesenchymal state, activating invasion and migration and attenuating proliferation. Snail1 induces migration and invasion in many different contexts through activation of a variety of factors including matrix metalloproteinases that enable cells to break the basal lamina and delaminate[55,56], and attenuates proliferation by repressing the *cyclin D* transcription[57]. Prrx1 also decreases cell proliferation and induces invasiveness[8,58]. Here, we show that some roles of Prrx1 are mediated by miR-15, such as promoting its own EMT mode by attenuating Snail1 expression. Interestingly, some miR-15f members have previously been associated with regulating cell proliferation, migration and

invasion[32–34], suggesting that in addition to Snail1, they have other targets that impinge into particular characteristics of the EMT process.

Overall, in this work, we describe a GRN that operates in development and in disease to regulate the activation of different EMT programs, providing a superimposed degree of controlled heterogeneity required for the EMT, a crucial and complex process, fundamental for organ formation and cancer progression.

## Methods

**Cell culture**. MDA-MB-231 (MDA231), MDA436, BT549, and SUM149PT human breast tumor cell lines, HEK293 human embryonic kidney and NIH3T3 mouse fibroblast cells were purchased from the ATCC (Virginia, USA) and ASTERAND BIOSCIENCE. MDA231, MDA436, and BT549 human tumor cell lines were cultured in DMEM:F12 HAM media (1:1), and SUM149PT cells in F12 HAM media, supplemented with 10% heat inactivated fetal bovine serum (FBS) (Sigma), 10 μg/ml insulin (Roche), 1% Gentamicin (Sigma), and 1% amphotericin (Sigma). HEK293 cells were cultured in DMEM supplemented with 10% heat inactivated FBS (Sigma) and 1% Gentamicin (Sigma) 1% amphotericin (Sigma). NIH3T3 cells were cultured in DMEM supplemented with 10% heat inactivated calf new-born serum (Sigma), 1% amphotericin (Sigma) and 1% Gentamicin (Sigma). Cells were kept at 37 °C and 5% $CO_2$, and the media was replaced every 2/3 days. HEK293 cells were passaged when they reached 80% confluency 1:10 every 48 h, while BT549 and MDA231, MDA436, and SUM149PT cells were passaged when reached 80% confluency 1:5 every 72 h. Cells were discarded after up to five consecutive passages and replaced by freshly thawed stocks. All cell lines were tested and confirmed negative for mycoplasma on a monthly bases at the host institution. All the cell lines were authenticated using STR profile by the Genetic Analysis Service at Miguel Hernandez University, Spain.

**Microarray analyses**. MDA231 human breast cancer cell line was transduced using a lentiviral system to stably express control GFP (MDA231-Control, MDA231-C) and PRRX1 (MDA231-PRRX1, MDA231-P)[8]. Three independent samples per condition from MDA231-C and MDA231-P cells, were hybridized to Human Gene 2.0 ST expression arrays (Affymetrix). Microarray data were analyzed using R and Bioconductor[59]. Expression data was preprocessed and normalized using the RMA algorithm[60], and differential expression analysis was performed using limma[61] *t* test to extract statistically significant changes between *PRRX1* and control samples. Accordingly, the genes that have a log2 fold change > 1.5 and corrected *p* value < 0.05 (Benjamini–Hochberg FDR) were considered as being differentially expressed.

**Plasmid constructs and interfering RNAs**. For overexpression studies, human PRRX1 ORF (ENSG00000116132) was cloned in the pBABE expression vector, and mouse Prrx1 ORF (ENSMUSG00000026586) was cloned in the pCDNA3.1 expression vector. Lentiviral vectors were generated via subcloning nuclear yellow fluorescent protein (nYFP) followed by P2A and mouse Prrx1-L open reading frame into pTRIPZ construct. P2A is a short peptide derived from porcine teschovirus-1 which mediates production of equimolar levels of Prrx1 protein sequence plus the nYFP reporter[62]. For RNA interference in BT549 cells, nYFP under hPGK promoter was subcloned into pLKO.1 construct containing PRRX1 specific shRNA (Dharmacon, RHS3979–201751764, Mature antisense: TTATTGGCTAGCATGGCTCTC)[8]. For RNA interference in MDA436 cells, specific siRNAs were used for SNAIL1 and PRRX1 (SNAIL1 siRNA (antisense): UGGCACUGGUACUUCUUGACAUCUGtt and PRRX1 siRNA (antisense): GAUGACUAAAUGGUAUUCCUtt). PRRX1 siRNA was obtained from Silencer® predesigned (Ambion). For miRNA overexpression studies, approximately 150 base pairs up- and downstream of the stem–loop sequences of each miRNA pair (obtained from www.mirbase.org) were cloned into the pcDNA3.1 vector. 3' UTRs of mouse, human, chicken and zebrafish *Snail1* were cloned downstream of Firefly luciferase ORF in pGL3 basic vector. Promoters of human and mouse *Mir-15-P1/2d* as well as human and mouse *Prrx1* plus human *PRRX1* upstream (enhancer-like; containing BS7) region (−7046 to −6439) were cloned upstream of the Firefly luciferase ORF in pGL3 basic vector, which contains a minimal promoter, considering cases to analyze DNA segments lacking promoter activity. For Snail1 reporter assay, mouse *Snail1* promoter cloned in pXP1 luciferase construct[63] was used. For mutagenesis, primers were designed using the online tool https://www.agilent.com/store/primerDesignProgram.jsp. PCR was performed using Pwo polymerase following manufacturer's instructions, and PCR products were treated with DpnI restriction enzyme, then transformed in bacteria to be amplified, extracted using Mini-prep and then sequenced to confirm the mutation/deletions. All constructs were generated using the primers described in Supplementary Table 3.

**Transfection of plasmids and interfering RNAs**. BT549 cells were transfected with either control pLKO.1 vector or PRRX1 specific shRNA[8] using Lipofectamin 3000 (Invitrogen) following the manufacturer's instructions. 500,000 cells were

seeded in 10 cm plates 24 h prior transfection. Five days post transfection YFP-positive cells were fluorescence-activated cell sorting (FACS) (BD FACSARIA III, USA) sorted directly collected into lysis solution from *mir*Vana™ miRNA Isolation Kit and subjected to RNA extraction. For LNA transfection, 250,000 BT549 cells were seeded in 6-well plates 24 h prior transfection, and 48 h after transfection cells were lysed for RNA extraction. HEK293 cells were transfected using the same procedure. MDA231 cells were transfected with pCDNA3.1 vectors for miRNA overexpression using Lipofectamin 3000 (Invitrogen) following the manufacturer's instructions. 200,000 cells were seeded in 6-well plates 24 h prior transfection, and 48 h after transfection cells were lysed for RNA extraction.

**Dual luciferase reporter assay**. HEK-293 and BT549 cells were cultured in 24-well plate 10 h before transfection. Cells were seeded in triplicate and transfected with 100 ng of the constructs containing sequences for promoters or 3′UTRs cloned in pGL3P vector, 50 ng of the pRL control vector (Renilla luciferase) and 200 ng of the pcDNA3.1 plasmid subcloned with different transgenes for pre-miRNA or Prrx1/Snail1 overexpression, using the Lipofectamin 2000 (Invitrogen). The media were refreshed 12 h after transfection. After 48 h, the cells were lysed using 5× lysis buffer (Promega), and the enzymatic activity was analyzed after adding Luciferase Assay and Stop&Glo reagents (Promega), to measure Firefly and Renilla successively, using Sirius Luminometer (Berthold, Germany).

**Lentiviral infection**. MDA231 cells were infected by pTRIPZ inducible lentiviral system containing nuclear nYFP-P2A-Prrx1 or control vectors with turbo RFP. The pool of infected cells was selected by treatment with puromycine for 48 h. Infected cells were seeded in 10 cm dishes, subjected to 2μg/ml doxycycline to induce the Tet-ON promoter (tetracycline-controlled transcriptional activation that is induced upon presence of antibiotic tetracycline or one of its derivatives, doxycycline) and FACS-sorted after 48 h for either YFP or RFP positive signals for Prrx1 overexpression or control cells, respectively. Cells were collected immediately after sorting in lysis solution from *mir*Vana™ miRNA Isolation Kit (Invitrogen).

**Total RNA extraction cDNA synthesis and qPCR analysis**. For gene expression assays, total RNA was extracted using the illustra RNAspin Mini isolation kit (GE Healthcare), following the manufacturers' instructions. For Reverse transcription cDNA synthesis, oligo (dT)$_{18}$ and random hexamer primers with the Maxima First Strand cDNA Synthesis kit (Thermo Scientific) were used, following the manufacturers' instructions. Primers used are listed in Supplementary Table 3. Quantitative RT-PCR was performed using Fast SYBR Green Mastermix (Applied Biosystems), in a Step One Plus machine (Applied Biosystems) according to the manufacturers' instructions. Relative levels of expression were calculated using the comparative Ct method normalized to the internal control TBP housekeeping gene.

For mature miRNA expression assays, total RNA enriched for small RNAs was extracted using *mir*Vana™ miRNA Isolation Kit (Invitrogen), following the manufacturers' instructions. Reverse transcription reactions were performed using TaqMan® MicroRNA Assay (Applied Biosystem) according to the manufacturers' instructions. For quantitative real-time retro-transcriptase PCR (qPCR), specific probes from TaqMan® MicroRNA Assay (Applied Biosystem) were used (Catalog # 4427975, hsa-miR-424-5p Assay ID: 000604 and hsa-miR-503–5p Assay ID: 001048). TaqMan® Universal Master Mix II, no UNG (Applied Biosystems) was used, in a Step One Plus machine (Applied Biosystems) according to the manufacturers' instructions. Relative levels of expression were calculated using the comparative Ct method normalized to the internal control U6 snRNA.

**ChIP assay**. Parental BT549 cells, BT549 cells transfected with SNAIL1-MYC and NIH-3T3 cells were used for ChIP assays for each indicated experiment. Cells were collected when at 80% confluency from 10 cm culture dish after trypsinization, and fixed in 1% PFA 15 min, and subsequently quenched with 0.125 M glycine for 15 min. Next, cells were lysed in 300 μl of lysis buffer (1% sodium dodecyl sulfate (SDS), 10 mM EDTA 50 mM Tris-HCl pH 8, protease inhibitor cocktail) on ice for 10 min. Lysates were sonicated four times for 15 min each in Bioruptor (H, 30″ on/ 30″ off). Sonicated lysates were then diluted to 5 ml with dilution solution (0.01% SDS, 1% Tx100, 2 mM EDTA, 20 mM Tris-HCl pH 8, 150 mM NaCl, protease inhibitor cocktail). Aliquots were de-crosslinked and ran in agarose gel to check for proper fragmentation. Samples were used for antibody incubation overnight at 4 °C in a rotating rotor. IgG was used as control for normalization and Histone H3 was used, at least in one of the replicates, to ensure efficient precipitation. Information and dilution for antibodies are listed in Table 1. Protein A coupled magnetic beads (Biorad) were blocked overnight with 0.05% bovine serum albumin (BSA), 2 μg/ml of salmon sperm DNA in dilution solution. Antibody and control fractions were then incubated with the magnetic beads for 4 h at 4 °C in a rotating rotor, and subsequently washed with Wash buffers (WB) 1 (0.1% SDS, 1% Tx100, 2 mM Tris-HCl pH 8, 150 mM NaCl), WB 2 (0.1% SDS, 1% Tx100, 2 mM Tris-HCl pH 8, 500 mM NaCl), WB 3 (1% NP40, 1% sodium deoxycholate, 10 mM Tris-HCl pH 8, 1 mM EDTA, 0.25 mM LiCl), and finally with WB4 (10 mM Tris-HCl pH8 and 1 mM EDTA). 10% CHELEX was added to the samples and then de-crosslinked at 95 °C for 10 min, Proteinase K treated (2 μg/ml) at 55 °C for 30 min and inactivated at 95 °C for 10 min. Supernatant was used for direct qPCR reaction (2.5 μl).

**RNA in situ hybridization**. Whole-mount RNA ISH[64] was carried out using digoxigenin-labeled probes synthesized from mouse, chicken and zebrafish Snail1 and Prrx1[12,41]. Mouse, chicken and zebrafish embryos were fixed in 4% PFA-DEPC O/N. Zebrafish embryos were then dechorionated in cold 4% PFA-DEPC. Embryos were washed in PBS 0.1% Tween 20 (PBS-T). Next, embryos were dehydrated through a series of increasing methanol concentrations in PBS-T (25, 50, 75, and 100%) and kept O/N at −20 °C. Then, they were rehydrated through methanol:PBS-T in reverse order and washed in PBS-T at the end. For fluorescent ISH, embryos were incubated in 1% hydrogen peroxide (H$_2$O$_2$) for 10 min, and then washed with PBS-T. Depending on species and developmental stage, embryos were treated with 10 μg/ml proteinase K in PBS-T between 3 and 6 min at room temperature. Then they were refixed with 4% PFA-DEPC, and washed. Embryos were then incubated with pre-hybridization solution (50% formamide, 5× SSC, 2% Boehringer blocking powder, 0.1% Tween 20, 50 μg/ml heparin, 1 mg/ml t-RNA, 1 mM EDTA, 0.1% CHAPS) at 60 °C, and then O/N upon refreshing prehybridization solution. Embryos were either used the next day for ISH or stored at −20 °C.

Prehybridized embryos were incubated with 1μg/ml of DIG or FLUO probes O/ N at 60 °C. The next day, embryos were washed several times first with 2X SSC, 0.1% CHAPS and then 0.2X SSC, 0.1% CHAPS, and then with KTBT washing buffer (50 mM Tris-HCl pH7.5, 150 mM NaCl, 10 mM KCl, 0.1% Triton X-100 in H$_2$O). After the washes, embryos were incubated in blocking solution (15% sheep serum, 0.7% Boehringer blocking solution, 0.1% Triton X-100 in KTBT) for 3 h at 4 °C. Then embryos were incubated with antibodies (1/1000 anti-DIG-AP, 1/500 anti-DIG-POD or anti-FLUO-POD, depending on the experiment) in blocking solution O/N at 4 °C. The whole next day, embryos were washed many times in KTBT buffer and kept O/N in KTBT at 4 °C.

For bright field ISH (chemical development of signal) embryos were washed in NTMT (100 mM Tris-HCl pH9.5, 59 mM MgCl$_2$, 100 mM NaCl, 0.1% Tween-20, 1 mM levamisole in H$_2$O) buffer three times prior to developing the signal. Embryos were then incubated with NTMT containing freshly added 3 μl NBT and 2.6 μl BCIP per 1 ml (developing solution), in the dark at RT until the color reaction develops. After obtaining the desired signal level in positive tissues, embryos were washed several times in KTBT and left O/N at 4 °C. After hybridization embryos were fixed in 4% PFA, washed in PBS and imaged in a Leica M125 dissecting scope with a Leica DFC 7000T digital camera (Leica, Wetzlar, Germany). Some embryos were embedded in paraffin or gelatin, and sections were obtained at 7 or 30 μm, respectively. Sections were photographed under a Leica DMR microscope (Leica, Wetzlar, Germany).

For double-fluorescent ISH, DIG, and FLUO labeled probes were mixed and added to the hybridization solution. After the washing steps with SSC buffer and blocking, embryos were incubated with the first antibody (anti-DIG-POD or anti-FLUO-POD). The next day, after two times washing with KTBT, embryos were incubated with Amplification solution (TSA® fluorescein detection kit, PerkinElmer) for 2 min to adjust the pH. Cy3 or FITC were added to the Amplification solution for developing red or green colors, respectively. The embryos were then incubated in the mix for 45 min in the dark at room temperature. Next, the samples were washed 5 times in KTBT, and incubated in 2% H$_2$O$_2$ for 2 h, then washed 5 times in KTBT. Embryos were then incubated in blocking solution for 3 h at 4 °C, prior to adding the second antibody (anti-DIG-POD or anti-FLUO-POD) in which they were incubated O/N at 4 °C. Information and dilution for antibodies are listed in Table 1. The next day, fluorescent developing procedures were repeated for the other color, plus stained with DAPI. After washing in KTBT, embryos were imaged with Olympus FV1200 confocal microscope and subjected to mosaic merge using Image J software, for pictures of whole-mounted embryos. Some embryos were embedded in 4% low-melting agarose and sectioned using a Leica VT1000S vibratome at 200 μm and subjected to confocal microscope imaging.

**miRNA in situ hybridization**. For mature miRNA ISH the mouse miR-322 (Mir-15-P1d), miR-503 (Mir-15-P2d), miR-15b (Mir-15-P1b), miR-16 (Mir-15-P2a/b) and scramble LNA probes, which were 5′–3′ DIG labeled, were purchased from Exiqon. ISH was performed on 10 μm frozen embryo sections, using the EDC method[65]. Dried cryo-sections were treated with 20 mg/ml proteinase K (pH7.4) in Tris-buffered saline (TBS) 1× (for 10× TBS, 69.6 g Tris, 87.6 g NaCl + 800 ml water, adjust pH to 7.6 and add water up to 1 L) for 20 min at RT, then washed 2 times in TBS and fixed in 4% PFA for 10 min at room temperature. Then, sections were washed once with TBS 0.2% glycine for 5 min, and washed twice with TBS.

Samples were then incubated twice for 10 min in Imidazole 0.13 M buffer (for 160 ml of buffer, 1.6 ml of 1-Methylimidazole was added to 130mlof water, pH was adjusted to 8 with HCl, and then 16 ml of NaCl 3 M was added and then water to final volume). Sections were fixed in 1-ethyl-3-(3-dimethylaminopropyl) carbodiimide (EDC) solution (176 μl of EDC (Sigma) was added to 10 ml of Imidazole buffer, then pH was readjusted to 8 by adding HCl) for 3 h at room temperature. Samples were washed once with TBS 0.2% glycine, and twice with TBS. Next, freshly prepared 0.1 M Triethanolamine (TEA), 05% acetic anhydride was added for 30 min at room temperature. After washing twice with TBS, the sections were prehybridized with hybrid-mix solution (50% formamide, 5× SSC, 5× Denhardt's solution (Applichem), 250 μg/ml yeast tRNA (Sigma), 500 μg/ml salmon sperm DNA (Sigma), 2% (w/v) Blocking Reagent (Roche), 0.1% 3-

**Table 1 Antibodies**

| Antibody | Concentration | Species and type | Provider (Cat #) |
|---|---|---|---|
| Prrx1 (for ChIP) | 1:200 | Rabbit polyclonal | Sigma (HPA051084) |
| IgG | 1:1000 | Rabbit | Diagenode (C15410206) |
| Myc tag | 1:500 | Goat pAb | Abcam (ab9132) |
| Prrx1 (for IF) | 1:100 | Tanaka lab[12] | |
| Snail1 (for IF in embryos and western blot) | 1:50 | Rabbit monoclonal | Cell Signaling (C15D3, #3879) |
| GFP | 1:500 | Chicken polyclonal | Aveslab (2BScientific, GFP-1020) |
| DIG-AP | 1:1000 | Fab fragments, sheep polyclonal | Roche (11093274910) |
| DIG-POD | 1:500 | Fab fragments, sheep polyclonal | Roche (11207733910) |
| FLUO-POD | 1:500 | Fab fragments, sheep polyclonal | Roche (11426346910) |
| β-actin | 1:2000 | Rabbit polyclonal | Abcam (ab8227) |
| Alexa Fluor 488 | 1:500 | Goat anti-rabbit | Invitrogen (A11008) |
| Alexa Fluor 568 | 1:500 | Goat anti-rabbit | Invitrogen (A11011) |
| Alexa Fluor 488 | 1:500 | Goat anti-chicken | Life technologies (A11039) |
| Alexa Fluor 568 | 1:500 | Goat anti-rat | Invitrogen (A11077) |
| Snail1 (for IF in cells) | 1:50 | Rat monoclonal | Cell signaling (SN9H2, #4719) |
| Histone H3 | 1:500 | Rabbit polyclonal | Abcam (ab1791) |

[(3-Cholamidopropyl) dimethylammonio]-1-propanesulfonate (CHAPs) (Sigma), 0.5% Tween) for 2 h at room temperature.

For hybridization, 4 pmol of DIG-labeled LNA probes were diluted in of hybrid-mix solution for each slide, and covered with Parafilm M (Sigma). The slides were kept in a sealed humidified chamber for at least 16 h at a temperature 20 °C below the melting temperature of the miRNA–LNA probes. Next day, sections were washed twice for 30 min in washing solution (50% formamide, 1X SSC, 0.1% Tween 20) at hybridization temperature, and then washed once with 0.2% SSC for 15 min at room temperature plus one wash with TBS 0.1% Tween 20. Slides were then incubated with 3% hydrogen peroxide, 0.1% Tween 20 in TBS for 30 min, and then washed 3 timed in TBS 0.1% Tween 20 at room temperature. Subsequently, sections were blocked by adding 0.5% Blocking Reagent (Roche), 10% sheep serum, 0.1% Tween 20 for 1 h at room temperature. Then, samples were incubated with anti-DIG antibody in blocking solution O/N at 4 °C. Information and dilution for antibodies are listed in Table 1. The next day, sections were washed 5 times with TBS 0.1% Tween 20, and incubated in NTMT buffer 3 times. Slides were next immersed in chambers containing developing solution (details in RNA ISH) at 37 °C until the color reaction developed. The sections were then washed several times in TBS and fixed in PFA before being photographed under a Leica DMR microscope (Leica, Wetzlar, Germany).

**Generation of the Tg(hsp68-GFP-DES1)54An mouse transgenic line.** The Tg (hsp68-GFP-DES1)54An line was generated after pronuclear injection of the vector-free reporter construct into FVB/NJ mouse blastocysts. The reporter construct contains mouse hsp68 (HSPA1A) minimal promoter (−872 to +1) driving eGFP followed by the SNAIL1 DES1 region (Downstream Enhancer SNAIL1). DES1 region contains the last 335 bp of SNAIL1 exon 3 and 1396 bp downstream of the gene. This region was cloned by PCR with the following primers: mmDES1-F 5′ GGATCCGCAGGGTGGTTACTGGACAC 3′ and mmDES1-R 5′ TGTCGACTCCTCCTCCCTCTCTGGAAT 3′. Founders were screened for the presence of GFP and they were mated to C57BL/6J mice to establish a line. Germ line transmission was tested both by PCR and GFP expression in E9.5 embryos. Two independent founders showed the same expression pattern and one of them (line 54) was backcrossed to C57BL/6J mice and used for further analysis.

**Mouse, chicken, and zebrafish embryo sections.** C57BL/6J wild-type mice were used. Embryos were staged as embryonic day (E) according to days post coitum. Wild-type Zebrafish strain AB were maintained at 28 °C under standard conditions, and the embryos were staged using zebrafish standard staging system[66]. Fertilized hen eggs were incubated in an humidified incubator at 37 °C, and embryos were staged based on Hamburger-Hamilton system[67].

For ISH to detect mRNA (see protocol above), mouse and chicken embryos were dehydrated through a series of increasing methanol:PBS-T series (25, 50, 75, and 100%) and twice in butanol, then embedded in paraffin O/N. Sectioning was performed in a Leica RM2245 microtome at 7 μm thickness.

For ISH to detect mature miRNA, mouse embryos were fixed in 4% PFA-DEPC (Diethyl pyrocarbonate) O/N. The next day embryos were washed twice with PBS, 0.1% Tween 20 before being embedded in 15% sucrose. Upon sinking, embryos were embedded in 30% sucrose, and after sinking they were kept in fresh 30% sucrose O/N. Embryos were then kept in a 1:1 mix of 30% sucrose:OCT for 30 min while rolling, before embedding in OCT. Embedded embryos were kept on dry ice and transferred to −80 °C before sectioning. OCT-embedded embryos were cryosectioned in a Slee MNT cryotome at 10 μm, dried for 2 h at room temperature before either being directly used for ISH (see protocol below) or stored at −80 °C.

Zebrafish embryos that were subjected to double-fluorescent ISH (see protocol below) were directly embedded in 4% low-melting agarose and sectioned using a Leica VT1000S vibratome at 100 μm, mounted using Dako fluorescent mounting medium and subjected to Confocal microscope imaging.

We affirm to have complied with all relevant ethical regulations for animal testing and research. All animal procedures were conducted in compliance with the European Community Council Directive (2010/63/EU) and Spanish legislation. The protocols were approved by the CSIC Ethical Committee and the Animal Welfare Committee at the Institute of Neurosciences, Alicante.

**miR-15 family sponge generation and injection in zebrafish.** We designed miR-15 sponge as inhibitors that can quench all members of the family[35], with the following sequence, ATCGTCGTAGTCATACCAAAAGCAATTCC (Supplementary Fig. 4k), in three consecutive repeats and cloned in pCS2 construct. Complementary sequences were chosen again conserved sequences among different family members, and few spacer nucleotides in between serve make bulges that can be recognized by the cell degradation machinery[35]. After linearization of the vector to synthesize the sponge transcripts followed by SV40 poly A, RNA was synthesized using the mMESSAGE mMACHINE SP6 Kit (Ambion) following manufacturer's indications prior injection in zebrafish.

For zebrafish embryo injections we first titrated the non-toxic effective dose of synthesized miR-15 family sponges to be injected safely. 100 ng of sponges were injected in the yolk of 1–2 cell stage embryos and incubated at 25 °C. At 20-somite stage, embryos were collected and fixed in 4% paraformaldehyde overnight at 4 °C for ISH or collected and lysed immediately for RNA extraction.

**Immunofluorescent (IF) staining.** Whole-mount mouse and chicken embryos were fixed in 4% PFA for 2 h at 4 °C. Then, embryos were dehydrated in a series of PBS 1% Triton x100 (PBS-T):methanol proportions (25, 50, 75, and 100%), then kept in 100% methanol O/N at −20 °C. Embryos where then rehydrated in the reverse order of PBS-T:methanol proportions. Antigen retrieval was performed by treating the mouse embryos with either 150 mM Tris-EDTA pH 9.0 (for Prrx1) or 10 mM Sodium Citrate pH 6.0 (for Snail1)-1% Triton x100 buffer at 70 °C for 20 min, following three washes with PBS-T. Embryos were blocked with 5% NGS 1% BSA 1% Tx-100 for 5 h or O/N at 4 °C and then incubated with the primary antibodies 48 h at 4 °C. After several washes for several hours in PBS-T, the embryos were incubated with the secondary antibodies and DAPI O/N at 4 °C. After washing the secondary antibody with PBS-T, the embryos were subjected to imaging. Pictures of whole-mount embryos were taken with an Olympus FV1200 confocal microscope with 10/20× objective and then subjected to mosaic merge using Image J software. Information and dilution for antibodies are listed in Table 1.

For cell lines IF, cells were cultured and treated on cover-slips in 6-well plates and collected at corresponding time points, fixed with 4% PFA for 15 min at room temperature, and washed three times with PBS. After washing fixed cells were either directly subjected to IF or stored at 4 °C. Cells were blocked with 5% NGS 1% BSA 0.2% Triton x-100 1 hour at room temperature and incubated with the primary antibodies O/N at 4 °C. After washing three times with PBS, cells were incubated with the secondary antibodies and DAPI 1 h at room temperature. Information and dilution for antibodies are listed in Table 1. After washing the secondary antibody with PBS, cells were photographed. Pictures were taken with Leica DMi8 microscope and HAMAMTSU C11440 digital camera.

**BMP treatment on zebrafish and chicken embryos.** Zebrafish embryos were collected and dechorionated at around 4-somite stage and treated with 200 μg/ml

of Human recombinant BMP4 protein (R&D Systems) which was added to the water, and kept at 25 °C. Embryos were collected after 30 min or 12 h and subjected to RNA extraction –qPCR. Collected embryos at different time points were subjected to ISH and IF for SNAIL1 and PRRX1. Chicken embryos were collected and cultured for 30 min prior BMP-soaked bead implantation. Human recombinant BMP2 protein (R&D Systems) was loaded onto heparin acrylic beads at a concentration of 0.2 mg/ml by soaking for 3 h at room temperature, and they were introduced into one half of the lateral plate mesoderm of chick embryos, close to the developing somites, at stage HH7, which were harvested at different time points (between 30 min to 10 h). Beads loaded with PBS were used as controls.

**Western blot**. E11.5 mouse embryos were dissected and lysed directly in RIPA buffer (25 mM Tris-HCl pH 7.6, 150 mM NaCl, 1% NP40, 1% sodium deoxy-cholate, 1% SDS and Proteinase inhibitor cocktail). Homozygous LacZ-Knock-in Prrx1 mutant embryos were identified among littermates by PCR genotyping and used for further analyses, together with wild-type embryos. Lysates were then homogenized followed by heating, sonication (6 rounds of 15 min, 30 s ON/30 s OFF, high). Samples were then loaded and subjected Coomassie Blue staining to determine the quality and concentration of the samples. Next, protein samples were loaded and migrated in 12% acrylamide gel and then transferred to PVDF western blotting membrane (Roche). The membranes were blocked with 10% milk and incubated with anti-Snail1 antibody overnight. Information and dilution for antibodies are listed in Table 1. The next day, membranes were extensively washed and incubated with secondary Peroxidase goat anti-Rabbit antibody, washed and developed using Chemiluminescent HRP Substrate (ImmobilonTM Western, Millipore). Pictures were obtained using Amersham Imager 680 (GE Healthcare). β-actin (Rabbit polyclonal, Abcam) was used as housekeeping protein for nor-malization. The uncropped and unprocessed scans of the blot are supplied in Supplementary Fig. 9.

**In silico analyses**. Single-cell RNA-seq public data: Processed datasets of publicly available data for single-cell RNA-seq were downloaded from NCBI Gene Expression Omnibus (GEO) database (https://www.ncbi.nlm.nih.gov/geo/). Data included single-cell RNA-seq from developing zebrafish and mouse embryos and different cancer types (Supplementary Table 1). Among analyzed single cells, the ones with no value for both Snail1 and Prrx1 were excluded. Hierarchical clustering was performed using GENE-E (version 3.0.215, Broad Institute, Inc.). Values were subjected to correlation analyses using Prism (GraphPad Softwares, Version 6.01, 2012) calculated based on Spearman r.

**Metanalysis of oncogenomic data from breast cancer patients**. To assess the putative correlations between the expression of SNAIL1, PRRX1 and mature miR-15 family members with OS in all subtypes ($n = 1403$), lymph-node positive patients ($n = 313$) or basal subtype ($n = 241$), all available data sets (E-TABM-43, GSE16716, GSE18728, GSE20194, GSE20271, GSE31448, GSE32646, GSE41998, GSE6532, GSE20711, GSE7390, GSE21653, GSE31519, GSE5327, GSE17907, E-MTAB-365, GSE37946, GSE2034, GSE2990, GSE17705, GSE1456, GSE12093, GSE9195, GSE45255, GSE20685, GSE12276, GSE2603, GSE16391, GSE42568, GSE11121, GSE3494, GSE16446, GSE4611, GSE26971, and GSE19615) were ana-lyzed. Kaplan–Meier plots were generated using http://kmplot.com[37,68]. The patient samples were grouped as either high or low for the expression of the genes of interest, and the auto-select best cut-off was chosen for computation over the entire data set.

**Statistical and data analysis**. Images were prepared using Adobe Photoshop and Adobe Illustrator CS6. Sample size was estimated using GPower 3.1, and values were set at $p = 0.05$ and beta = 0.8. All statistical analyses were performed using Microsoft Excel 2013 and Prism (GraphPad Softwares, Version 6.01, 2012). For reporter assays and qRT–PCR experiments, the corresponding treatments were compared with controls using Student's two-tailored t test or One-way ANOVA with Bonferroni's multiple comparison test. Spearman r was used for correlation test of single-cell RNA-seq data. All bar graphs represent mean + SEM (Standard Error of the Mean). Statistical significances were as follows: *$P \leq 0.05$, **$P \leq 0.01$ and ***$P \leq 0.001$.

**Reporting summary**. Further information on research design is available in the Nature Research Reporting Summary linked to this article.

## Data availability
The microarray data generated in this study, as well as public scRNA-seq data datasets analyzed during the current study are available in the Gene Expression Omnibus (GEO) repository under the following accession numbers that are also listed in Supplementary Table 1. Microarray in MDA231 cells: GSE138078, Zebrafish embryos: GSM3067194, mouse embryo: GSE87038, head and neck cancer patients: GSE103322, and breast cancer patients: GSE75688. The source data underlying Fig. 1c, f–h, 2a–f, 3b–f, 4a–e and h, and 6a, and Supplementary Figs. 1f, g, 2a–f, 3a and c–i, 4a and g–j, 5c and e, and 7b are provided as a Source Data file.

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

## Acknowledgements

We thank Sonia Vega for her help and support in managing cell lines, Verona Villar Cerviño for advice with imaging analysis, Cristina López Blau, Diana Abad Bataller and Sandra Moreno Valverde for helpful technical support, and other lab members for continuous and helpful discussions. Also we thank Antonio Caler Escribano for technical help in the FACS/Omics facility. This work was supported by grants from the Spanish Ministries of Economy and Competitiveness (MINECO BFU2014-53128-R), of Science, Innovation and Universities (MICIU RTI2018-096501-B-I00), Generalitat Valenciana (2017/150) and the European Research Council (ERC AdG 322694) to M.A.N., who also acknowledges financial support from the Spanish State Research Agency, through the "Severo Ochoa" Program for Centres of Excellence in R&D (SEV-2017–0273). H.F. was recipient of PhD student scholarships Santiago Grisolia from Generalitat Valenciana (GRISOLIA/2014/004). L.R. was holder of a Juan de la Cierva Formación postdoctoral scholarship and F.G.A. is recipient of a PhD student scholarship (FPI), both from MINECO.

## Author contributions

H.F. performed the majority of experiments, analyzed and interpreted the data, and wrote the paper. L.R. injected and treated the zebrafish embryos, did the ChIP assay in Supplementary Fig. 2c, e and contributed to ISHs in Figs. 1, 3 and 5. K.K.Y. contributed to conditional in vitro gain of function of Prrx1 (Fig. 3c) and expression analyses in Supplementary Fig. 1e, plus knockdown experiments in Supplementary Fig. 1g. O.H.O. performed the MicroArray experiments (Fig. 3a), BMP treatment of chicken embryos and ISHs in Fig. 1d. F.G.A. performed western-blots in Fig. 5i and LacZ staining (Supplementary Fig. 6a). A.A. analyzed the MicroArray (Fig. 3a) and did some in silico analysis. J.G. conceived, designed, and generated the mouse transgenic line. M.A.N. conceived the project, designed experiments, interpreted the data, wrote the paper, and secured funding.

## Competing interests

The authors declare no competing interests.

## Additional information

**Peer review information** *Nature Communications* thanks Gregory Goodall and other, anonymous, reviewers for their contributions to the peer review information. Peer review reports are available.

