## [Peer Review File · Nature Communications]

Reviewers' comments:

Reviewer #1 (Remarks to the Author):

Fazilaty et al. show that Snail1 and Prrx1 are expressed in complementary fashion in cells that undergo EMT in developing embryos of fish, chicken and mice, and are negatively correlated in cell lines and cancers. They show in cell culture that this reciprocal pattern is set up by a double negative feedback loop, in which Snail transcriptionally represses Prrx1, while Prrx1 activates its own promoter and indirectly represses Snail by its induction of miR-424 and miR-503, members of the miR-15 family of microRNAs. They propose the two different modes of repression allow initial expression of Snail1 with slower induction of Prrx1 eventually supplanting the Snail to allow a different EMT profile in cells where the Prrx1 is induced.

The use of multiple in vivo models combined with in vitro experiments build a strong case for the proposed mechanism, and the discussion of potential consequences in both the developmental and cancer contexts is thoughtful and interesting. I have only minor suggestions for improvement.

I am rather surprised the repression of luciferase reporter genes by miR-15-P1/2d is so strong for all 4 species of Snail 3'UTR tested, given the seed match is only a 6-mer, which usually only gives weak suppression. In human the binding is augmented by some good complementarity to the 3' end of the miRNAs, but it is not obvious this is the case for the other species tested. Is there augmenting binding also in the other species?

It would be useful to include scatter plots for the correlation data in Fig 1 f,g,h and for each cell line in Supp Fig 1f, to provide easily recognisable indications of the relationship between Snail1 and Prrx1 across the different levels of expression.

The conclusion that Prrx1 does not directly regulate Snail1 transcription (and hence regulates indirectly) would be even stronger if the effect of Prrx1 on a Snail promoter-luciferase reporter was tested, rather than relying solely on lack of detection of binding in ChIP assays of small selected regions in the promoter.

The axis label for Fig 3d and Supp Fig 3c should say "Relative pre-miR levels", not "Relative mRNA levels".

Reviewer #2 (Remarks to the Author):

Fazilaty and co-workers report the intriguing reciprocal regulation of the transcription factors Snail1 and Prrx1 during an EMT in embryonic development and in malignant tumor progression. Using an elegant combination of experiments with Zebrafish, chick and mouse embryos and cell and molecular biology and biochemistry experiments with murine and human cancer cells in vitro and with cancer cell transplantations in vivo, they document that Snail1 directly represses Prrx1 transcription, while Prrx1 via the induction of miR-15 family members represses the expression of Snail1. Altogether the data provides several lines of evidence for the importance of this regulatory network in the general process of an EMT and thus pinpoints a critical rheostat of the regulation of an EMT in physiology and disease.

The manuscript is presented in a concise and refreshingly clear manner. The experiments are in general thoughtfully designed and well-controlled and the conclusions drawn by the authors are widely supported by the experimental data presented. The Discussion nicely embeds the new insights into current knowledge and points to important future directions.

Yet, there are a few points that are not convincingly clear and should be considered:

1. Introduction: "drives selection of EMT mode". What is the definition of an EMT mode? This needs some explanation.
2. The term "complementary" is used many times for the converse regulation of the two transcription factors. Would not "reciprocal" or "mutually exclusive" better define the actual functional connection between the transcription factors?
3. Figure S1 e and f: It would be highly informative if the migratory and invasive potential and the expression of general EMT markers would be correlated with the expression of Snail1 and Prx1 in the cancer cell lines analyzed.
4. Figure 2: Is changing the relative expression levels of Snail1 and Prx1 in MDA 436 cells by loss and gain of function experiments having an effect on the endogenous levels, for example by the forced expression of a miR-15-insensitive construct encoding for Snail1? Do protein levels change accordingly? The half-life of the proteins may play an important role in their rheostat function. It is noted that the Prx1 mRNA levels are reduced only by 60% upon overexpression of Snail1 in BT549 cells. This seems rather weak, in particular when considering the self-regulatory loop of Prx1 on its own expression. Hence, the correlation between mRNA and protein levels may be important.
5. Figure 2b and e: Have the various binding sites (BS) been mutated in the reporter constructs to show specificity?
6. Figure 3g: Higher magnification of the microscopic pictures should be shown to illustrate the cellular co-localization of Prx1 and the miR-15 family members.
7. Figure S4i: this result should be included into Figure 4f, including mentioning the number of injected and affected embryos.
8. Figure 5: Snail1 apparently has a much higher significance in its prognostic value as Prx1? How is this explained? May other regulatory circuits impinging on Snail1 expression play a role as well, for example miR-34?
9. Figure 5: It would be of high interest to see the Kaplan-Meier curves for the different subtypes of breast cancer and also for metastasis-free survival.
10. Figure 6a and b: The elegant tracing experiment still leaves an open question with regard to the sequential activation of the transcription factors dependent on their reciprocal regulation: Since Snail comes on first, why is Prx1 expression increasing at later times (5 hours). To explain such temporal behavior, one would have to argue and experimentally test whether the inductive effect of BMP2 differs between the expression of Snail1 and Prx1 in a manner of actual transcriptional activation or in a temporal manner. Conversely, with the increase of Prx1, the expression of miR-15 should increase and the expression of Snail1 should decrease, a notion not obvious in the results presented in the figure. Hence, there remains a question as to how BMP2 can overcome the repression of Prx1 by Snail1 to upregulated Prx1 expression? As already mentioned above, a consideration of protein levels may be instructive. Moreover, in addition to the cell autonomous pathways, as shown by the cell tracing experiments, are there paracrine effects involved as well, such as the influence of neighboring cells by cell adhesion?
11. For a full understanding of the specific roles of Snail1 and Prx1 in the regulation of EMT the identification of their direct target genes by ChIP-Seq and RNA Seq would be important, in particular to identify the genes and regulatory circuits underlying EMT-induced cell plasticity suspected to be found in partial or hybrid EMT stages.

12. Overall English style and Grammar will benefit from additional proofreading.

Reviewer #3 (Remarks to the Author):

Although EMT is a very crucial process in development as well as in several human diseases, the function of individual EMT-TF and how they are regulated is still poorly understood. In this manuscript the authors identified a novel gene regulatory network that involves two EMT-TFs, Snail and Prrx1. They elegantly and convincingly show that both are expressed in a complementary manner by establishment of a double negative feedback loop that involves direct repression of Prrx1 by Snail, Prrx1-dependent activation of miR-15 transcription and Snail attenuation by binding miR-15 family members to the 3'UTR of Snail. The analyses are done in a very accurate manner by using high level state of the art techniques in mouse, chicken and zebrafish embryos. The experiments provide very robust results that support the conclusions. I truly recommend this manuscript for publication in Nature Communications. I listed some points that need to be addressed to improve the manuscript prior to publication:

1. All ChIP experiments are done in a setting with exogenous overexpression of Snail and Prrx1. Especially, since Prrx1 is precipitated with an antibody that detects the unmodified/untagged protein, a ChIP should be included using endogenously expressed proteins. Moreover, the ChIP results should be presented as percentage of input to apply to ChIP standards. The position of NC regions that were used as negative controls should be provided.
2. Although the implantation of BMP2 coated beads is rather unphysiological, I was puzzled by the observation that both, Snail and Prrx1, are upregulated in the chick embryo upon bead implantation. Since Snail is induced very early, the double-negative feedback loop should prevent Prrx1 activation. How is this discrepancy explained?
3. Fig. 2D and text: The authors claim that binding to BS7 is significant. However, this is misleading, as the lack of significance for binding sites 2-6 can be due to the fact that only two data points are shown for these sites. By increasing the n number, a similar level of significance could be present here as well. It Why are not all BS's analyzed in each ChIP experiment (the figure legends states that n = 4-6)?

Reviewer #1 (Remarks to the Author):

We would like to thank this reviewer for the constructive review, for the interest in our work, and the acknowledgement of the use of multiple *in vitro* and *in vivo* models. In the revised version, the new text is highlighted in blue (including Figure legends and Supplementary Material) to better identify the changes made to the previous version.

Fazilaty et al. show that Snail1 and Prrx1 are expressed in complementary fashion in cells that undergo EMT in developing embryos of fish, chicken and mice, and are negatively correlated in cell lines and cancers. They show in cell culture that this reciprocal pattern is set up by a double negative feedback loop, in which Snail transcriptionally represses Prrx1, while Prrx1 activates its own promoter and indirectly represses Snail by its induction of miR-424 and miR-503, members of the miR-15 family of microRNAs. They propose the two different modes of repression allow initial expression of Snail1 with slower induction of Prrx1 eventually supplanting the Snail to allow a different EMT profile in cells where the Prrx1 is induced.

The use of multiple *in vivo* models combined with *in vitro* experiments build a strong case for the proposed mechanism, and the discussion of potential consequences in both the developmental and cancer contexts is thoughtful and interesting. I have only minor suggestions for improvement.

I am rather surprised the repression of luciferase reporter genes by miR-15-P1/2d is so strong for all 4 species of Snail 3'UTR tested, given the seed match is only a 6-mer, which usually only gives weak suppression. In human the binding is augmented by some good complementarity to the 3' end of the miRNAs, but it is not obvious this is the case for the other species tested. Is there augmenting binding also in the other species?

Thanks for the suggestion. The same *in-silico* prediction for miR hybridization we had included on human *SNAIL1* 3'UTR (shown in Supplementary Figure 4b) has been done for miRs from mouse, chicken and fish. We find that all miR-15 family members are predicted to bind to *Snail1* 3'UTR not only in the seed region but also in the 3' end of the miRNAs (see new Supplementary Figure 4d-f).

It would be useful to include scatter plots for the correlation data in Fig 1 f,g,h and for each cell line in Supp Fig 1f, to provide easily recognisable indications of the relationship between Snail1 and Prrx1 across the different levels of expression.

Thanks for this suggestion. We had already attempted to do so but in Fig. 1f-h, due to the fact that there are many cells with 0 value for one of the gene transcripts, the plot is not as visually informative as the heatmap. In the case of Supp. Fig 1f, for MDA436 cells, as the majority of cells express the two genes at similar intermediate levels, the scatter plot is not informative either. This experiment exemplifies one of the conclusions of this work and the described GRN. Coexpression at low/moderate levels is possible, whereas the coexistence of Snail1 and Prrx1 at high levels in cells does not occur. See also our response to point 3 raised by Reviewer 2.

The conclusion that Prrx1 does not directly regulate Snail1 transcription (and hence regulates indirectly) would be even stronger if the effect of Prrx1 on a Snail promoter-luciferase reporter was tested, rather than relying solely on lack of detection of binding in ChIP assays of small selected regions in the promoter.

Thanks again. In response to this, we now include Luciferase assays to show Prrx1 transfection does not change the Luciferase activity downstream of *Snail1* promoter, confirming that Prrx1 does not directly repress *Snail1* transcription (see new Supplementary Figure 2f).

The axis label for Fig 3d and Supp Fig 3c should say “Relative pre-miR levels”, not “Relative mRNA levels” We apologize for the mistake. However, it could not be changed to “pre-miR” as the graphs also include *PRRX1* expression. We have corrected it to “Relative RNA levels”. Thanks.

Reviewer #2 (Remarks to the Author):

We thank very much this reviewer for the thorough and helpful review, the suggestions, the interest in our work and for the comments on the conclusions and on the discussion as we have integrated previous and new findings into a common conceptual frame. In the revised version, the new text is highlighted in blue (including Figure legends and Supplementary Material) to better identify the changes made to the previous version.

Fazilaty and co-workers report the intriguing reciprocal regulation of the transcription factors Snail1 and Prrx1 during an EMT in embryonic development and in malignant tumor progression. Using an elegant combination of experiments with Zebrafish, chick and mouse embryos and cell and molecular biology and biochemistry experiments with murine and human cancer cells in vitro and with cancer cell transplantations in vivo, they document that Snail1 directly represses Prrx1 transcription, while Prrx1 via the induction of miR-15 family members represses the expression of Snail1. Altogether the data provides several lines of evidence for the importance of this regulatory network in the general process of an EMT and thus pinpoints a critical rheostat of the regulation of an EMT in physiology and disease.

The manuscript is presented in a concise and refreshingly clear manner. The experiments are in general thoughtfully designed and well-controlled and the conclusions drawn by the authors are widely supported by the experimental data presented. The Discussion nicely embeds the new insights into current knowledge and points to important future directions.

Yet, there are a few points that are not convincingly clear and should be considered:

1. Introduction: “drives selection of EMT mode”. What is the definition of an EMT mode? This needs some explanation.

Thanks for pointing out that we were not clear in the previous version. We have included two additional sentences in the introduction.

2. The term “complementary” is used many times for the converse regulation of the two transcription factors. Would not “reciprocal” or “mutually exclusive” better define the actual functional connection between the transcription factors?

We appreciate the comment, and we had already thought about the best term to use. We believe that complementary is a better option, because there are cells that express both (at low/moderate levels, still compatible with the network), indicating that this is not a mutually exclusive situation.

3. Figure S1 e and f: It would be highly informative if the migratory and invasive potential and the expression of general EMT markers would be correlated with the expression of Snail1 and Prrx1 in the cancer cell lines analyzed.

All three cell lines are triple-negative/basal type and show mesenchymal properties with migratory and invasive potential. The main differences are that SUM and MDA cells are metastatic, while BT549 is not; and the two first cell lines have cancer stem cell-like properties, while BT549 cells do not. Importantly, the latter gain both stem cell-like properties and metastatic ability upon downregulation of PRRX1, as described in Ocaña et al., *Cancer Cell*, 2012. A few sentences and references are added in the main text addressing these issues (page 4).

4. Figure 2: Is changing the relative expression levels of Snail1 and Prrx1 in MDA 436 cells by loss and gain of function experiments having an effect on the endogenous levels, for example by the forced expression of a miR-15-insensitive construct encoding for Snail1? Do protein levels change accordingly? The half-life of the proteins may play an important role in their rheostat function. It is noted that the Prrx1 mRNA levels are reduced only by 60% upon overexpression of Snail1 in BT549 cells. This seems rather weak, in particular when considering the self-regulatory loop of Prrx1 on its own expression. Hence, the correlation between mRNA and protein levels may be important.

Thanks very much for the suggestion. We have added new data as Supplementary Figure 1g. We have downregulated either SNAIL1 or PRRX1 using siRNA and examine the impact on this acute downregulation in MDA436 cells, to test how the network responds in a cell line that has stable low/moderate levels of both transcription factors. We find an increased expression of *PRRX1* and *SNAIL1*, respectively, indicating that these cells respond in a way predicted by the GRN proposed here.

5. Figure 2b and e: Have the various bindings sites (BS) been mutated in the reporter constructs to show specificity?

Thanks very much for this suggestion. We should have performed this experiment already in the previous version. We find that the sole deletion of BS2 is sufficient to abolish the repression of *PRRX1* transcription by *SNAIL1*, and also that deletion of BS7 abolish *PRRX1* transcriptional autoactivation (see Figures 2b and e).

6. Figure 3g: Higher magnification of the microscopic pictures should be shown to illustrate the cellular co-localization of Prrx1 and the miR-15 family members.

We thank the reviewer for this suggestion. Unfortunately, this experiment is not a double in situ hybridization, which would not work. Therefore, cellular co-localization of Prrx1 and miRs cannot be addressed. It was meant to show that miRs of the 15 family and Prrx1 are expressed in the same regions and at the same time in the mouse embryo.

7. Figure S4i: this result should be included into Figure 4f, including mentioning the number of injected and affected embryos.

If possible, we would like to keep it in the supplementary figure (now Suppl. Fig. 4I), because this image does not represent the majority of embryos.

8. Figure 5: Snail1 apparently has a much higher significance in its prognostic value as Prxx1? How is this explained? May other regulatory circuits impinging on Snail1 expression play a role as well, for example miR-34?

Thanks for spotting this. The higher prognostic value of SNAIL1 observed in Lymph-node positive patients, is not sustained when we examined all patients with breast cancer. We have decided to move these data from Suppl. Fig. 6 to Fig. 5a. In addition, a sentence addressing the possibility of additional networks impinging on Snail expression has been added in the discussion section (page 14).

9. Figure 5: It would be of high interest to see the Kaplan-Meyer curves for the different subtypes of breast cancer and also for metastasis-free survival.

[REDACTED]

10. Figure 6a and b: The elegant tracing experiment still leaves an open question with regard to the sequential activation of the transcription factors dependent on their reciprocal regulation: Since Snail comes on first, why is Prxx1 expression increasing at later times (5 hours). To explain such temporal behavior, one would have to argue and experimentally test whether the inductive effect of BMP2 differs between the expression of Snail1 and Prxx1 in a manner of actual transcriptional activation or in a temporal manner. Conversely, with the increase of Prxx1, the expression of miR-15 should increase and

the expression of Snail1 should decrease, a notion not obvious in the results presented in the figure. Hence, there remains a question as to how BMP2 can overcome the repression of Prx1 by Snail1 to upregulated Prx1 expression? As already mentioned above, a consideration of protein levels may be instructive. Moreover, in addition to the cell autonomous pathways, as shown by the cell tracing experiments, are there paracrine effects involved as well, such as the influence of neighboring cells by cell adhesion?

We thank very much the reviewer for this excellent suggestion. Indeed, *in vivo* evidence of *Snail1* downregulation after the onset of BMP-induced *Prx1* activation was missing. The experiment to answer this question is included in a modified version of Figure 6b and in Supplementary Fig. 7b, c. We now show that 10h after BMP administration, and as expected in our model, *SNAIL1* transcription is downregulated, while *PRRX1* is still highly induced in the somites close to the bead. Protein expression quantification also supports that *PRRX1* protein is upregulated in the somites close to the bead 10h after bead implantation, when *SNAIL1* transcription is downregulated. Regarding *SNAIL1* protein, when left and right somites are compared at different time points, a significant increase is observed 5h after bead implantation (Supplementary Fig. 7c). This may also explain why although *SNAIL1* transcription is rapidly upregulated, the gradual increase in protein levels allows *PRRX1* to be upregulated in response to BMP. Later on, *PRRX1* high levels will outcompete *SNAIL1* transcription.

In addition, to reinforce this concept, we have included a new experiment in which we have incubated zebrafish embryos with a solution containing BMP. We find that a similar temporal response also occurs in the fish (see Figure 6a). In this case, we have also examined the expression of miRs.

11. For a full understanding of the specific roles of Snail1 and Prx1 in the regulation of EMT the identification of their direct target genes by ChIP-Seq and RNA Seq would be important, in particular to identify the genes and regulatory circuits underlying EMT-induced cell plasticity suspected to be found in partial or hybrid EMT stages.

We fully agree with the reviewer. However, we believe that the identification, validation and corresponding *in vivo* functional analyses of the most promising candidates constitutes a sufficient additional body of evidence for an independent study.

12. Overall English style and Grammar will benefit from additional proofreading.

Thanks for the suggestion. We have double-checked mistakes

Reviewer #3 (Remarks to the Author):

We thank very much this reviewer for the constructive review, the suggestions and for the comments on the robustness and high quality of our experimental approaches and the firm support of our work as suitable for Nature Communications. In the revised version, the new text is highlighted in blue (including Figure legends and Supplementary Material) to better identify the changes made to the previous version.

Although EMT is a very crucial process in development as well as in several human diseases, the function of individual EMT-TF and how they are regulated is still poorly understood. In this manuscript the

authors identified a novel gene regulatory network that involves two EMT-TFs, Snail and Prrx1. They elegantly and convincingly show that both are expressed in a complementary manner by establishment of a double negative feedback loop that involves direct repression of Prrx1 by Snail, Prrx1-dependent activation of miR-15 transcription and Snail attenuation by binding miR-15 family members to the 3'UTR of Snail. The analyses are done in a very accurate manner by using high level state of the art techniques in mouse, chicken and zebrafish embryos. The experiments provide very robust results that support the conclusions. I truly recommend this manuscript for publication in Nature Communications. I listed some points that need to be addressed to improve the manuscript prior to publication:

1. All ChIP experiments are done in a setting with exogenous overexpression of Snail and Prrx1. Especially, since Prrx1 is precipitated with an antibody that detects the unmodified/untagged protein, a ChIP should be included using endogenously expressed proteins.

We are sorry that we were not clear in the description of some of our experiments. All ChIP experiments for Prrx1 both in human and mouse cells have been performed with endogenous protein. In the case of Snai1, we had to overexpress a Myc-tagged Snail1 version for ChIP analyses because the anti-Snail1 antibodies are suitable for ChIP.

Moreover, the ChIP results should be presented as percentage of input to apply to ChIP standards. The position of NC regions that were used as negative controls should be provided.

Thanks for the suggestion. All the ChIP experiments are re-calculated as the percentage of input. Also, NC regions are added in the figure legends.

2. Although the implantation of BMP2 coated beads is rather unphysiological, I was puzzled by the observation that both, Snail and Prrx1, are upregulated in the chick embryo upon bead implantation. Since Snail is induced very early, the double-negative feedback loop should prevent Prrx1 activation. How is this discrepancy explained?

We thank very much the reviewer for spotting this apparent contradiction. In response to this and also to the suggestion made by reviewer 2 (point #10), we have now included new data in Fig. 6 and Supplementary Fig. 7. We now provide *in vivo* evidence both in the fish and the chick of *Snail1* downregulation after the onset of BMP-induced Prrx1 activation (Figure 6a, b and Supplementary Fig. 7). As expected from our model, *SNAIL1* transcription is rapidly upregulated, the gradual increase in protein levels allows *PRRX1* to be upregulated in response to BMP. Later on, *PRRX1* high levels will outcompete *SNAIL1* transcription (see also answer to reviewer 2 and the description of these experiments in page 11).

3. Fig. 2D and text: The authors claim that binding to BS7 is significant. However, this is misleading, as the lack of significance for binding sites 2-6 can be due to the fact that only two data points are shown for these sites. By increasing the n number, a similar level of significance could be present here as well. It Why are not all BS's analyzed in each ChIP experiment (the figure legends states that n = 4-6)?

We thank the reviewer very much for the suggestion. We agree that the number of biological replicates was not sufficient in the previous version. We have repeated the experiments, and we are pleased to confirm that some BS now show significant enrichment (See Fig. 2).

REVIEWERS' COMMENTS:

Reviewer #1 (Remarks to the Author):

The points raised in my previous review of the manuscript have been satisfactorily addressed.

Reviewer #2 (Remarks to the Author):

The authors have appropriately responded to all of the reviewers' comments. They have added relevant experimental data and revised the manuscript accordingly to clarify open or unclear points.

There are only a few mishaps and typos:

Page 9, upper paragraph, last sentence: It should say Suppl. Figure 4g-j instead of 4e-g.

Page 9 middle paragraph, first sentence: it should say Suppl. Figure 4k instead of 4h.

Page 10, last paragraph: Therefore, we examined....

Reviewer #3 (Remarks to the Author):

All points of my review were sufficiently addressed. All new data even further support the conclusions and I would like to congratulate the authors for this nice piece of work!

RESPONSE TO REVIEWERS' COMMENTS:

First of all, we would like to mention that we are very pleased with the constructive reviews throughout the process and appreciate how their comments have helped us improve our study.

Reviewer #1 (Remarks to the Author):

The points raised in my previous review of the manuscript have been satisfactorily addressed.

We would like to thank this reviewer for the constructive review, for the interest in our work, and the acknowledgement of addressing the points raised.

Reviewer #2 (Remarks to the Author):

The authors have appropriately responded to all of the reviewers' comments. They have added relevant experimental data and revised the manuscript accordingly to clarify open or unclear points.

There are only a few mishaps and typos:

Page 9, upper paragraph, last sentence: It should say Suppl. Figure 4g-j instead of 4e-g.

Page 9 middle paragraph, first sentence: it should say Suppl. Figure 4k instead of 4h.

Page 10, last paragraph: Therefore, we examined....

We thank very much this reviewer for the constructive, thorough and helpful review and the interest in our work. The few mishaps and typos are all addressed in the revised version.

Reviewer #3 (Remarks to the Author):

All points of my review were sufficiently addressed. All new data even further support the conclusions and I would like to congratulate the authors for this nice piece of work!

We thank very much this reviewer for the constructive review, the very helpful suggestions and the comments. We also thank the reviewer for their kind congratulations message.